# On some provably correct cases of variational inference for topic models

**Pranjal Awasthi**
Department of Computer Science
Rutgers University
New Brunswick, NJ 08901
pranjal.awasthi@rutgers.edu

**Andrej Risteski**
Department of Computer Science
Princeton University
Princeton, NJ 08540
risteski@cs.princeton.edu

## Abstract

Variational inference is an efficient, popular heuristic used in the context of latent variable models. We provide the first analysis of instances where variational inference algorithms converge to the global optimum, in the setting of topic models. Our initializations are natural, one of them being used in LDA-c, the most popular implementation of variational inference. In addition to providing intuition into why this heuristic might work in practice, the multiplicative, rather than additive nature of the variational inference updates forces us to use non-standard proof arguments, which we believe might be of general theoretical interest.

## 1 Introduction

Over the last few years, heuristics for non-convex optimization have emerged as one of the most fascinating phenomena for theoretical study in machine learning. Methods like alternating minimization, EM, variational inference and the like enjoy immense popularity among ML practitioners, and with good reason: they're vastly more efficient than alternate available methods like convex relaxations, and are usually easily modified to suite different applications.

Theoretical understanding however is sparse and we know of very few instances where these methods come with formal guarantees. Among more classical results in this direction are the analyses of Lloyd's algorithm for K-means, which is very closely related to the EM algorithm for mixtures of Gaussians [20], [13], [14]. The recent work of [9] also characterizes global convergence properties of the EM algorithm for more general settings. Another line of recent work has focused on a different heuristic called *alternating minimization* in the context of dictionary learning. [1], [6] prove that with appropriate initialization, alternating minimization can provably recover the ground truth. [22] have proven similar results in the context of phase retreival.

Another popular heuristic which has so far eluded such attempts is known as *variational inference* [19]. We provide the first characterization of global convergence of variational inference based algorithms for topic models [12]. We show that under natural assumptions on the topic-word matrix and the topic priors, along with natural initialization, variational inference converges to the parameters of the underlying ground truth model. To prove our result we need to overcome a number of technical hurdles which are unique to the nature of variational inference. Firstly, the difficulty in analyzing alternating minimization methods for dictionary learning is alleviated by the fact that one can come up with closed form expressions for the updates of the dictionary matrix. We do not have this luxury. Second, the "norm" in which variational inference naturally operates is KL divergence, which can be difficult to work with. We stress that the focus of this work is not to identify new instances of topic modeling that were previously not known to be efficiently solvable, but rather providing understanding about the behaviour of variational inference, the defacto method for learning and inference in the context of topic models.

## 2   Latent variable models and EM

We briefly review EM and variational methods. The setting is latent variable models, where the observations $X_i$ are generated according to a distribution $P(X_i|\theta) = P(Z_i|\theta)P(X_i|Z_i,\theta)$ where $\theta$ are parameters of the models, and $Z_i$ is a latent variable. Given the observations $X_i$, a common task is to find the max likelihood value of the parameter $\theta$: $\mathrm{argmax}_\theta \sum_i \log(P(X_i|\theta))$.

The EM algorithm is an iterative method to achieve this, dating all the way back to [15] and [24] in the 70s. In the above framework it can be formulated as the following procedure, maintaining estimates $\theta^t, \tilde{P}^t(Z)$ of the model parameters and the posterior distribution over the hidden variables: In the E-step, we compute the distribution $\tilde{P}^t(Z) = P(Z|X,\theta^t)$. In the M-step, we set $\theta^{t+1} = \mathrm{argmax}_\theta \sum_i E_{\tilde{P}^t}[\log P(X_i, Z_i|\theta)]$. Sometimes even the above two steps may not be computationally feasible, in which case they can be relaxed by choosing a family of simple distributions $F$, and performing the following updates. In the variational E-step, we compute the distribution $\tilde{P}^t(Z) = \min_{P^t \in F} KL(P^t(Z)||P(Z|X,\theta^t))$. In the M-step, we set $\theta^{t+1} = \mathrm{argmax}_\theta \sum_i E_{\tilde{P}^t}[\log P(X_i, Z_i|\theta)]$. By picking the family $F$ appropriately, it's often possible to make both steps above run in polynomial time. As expected, none of the above two families of approximations, come with any provable global convergence guarantees. With EM, the problem is ensuring that one does not get stuck in a local optimum. With variational EM, additionally, we are faced with the issue of in principle not even exploring the entire space of solutions.

## 3   Topic models and prior work

We focus on a particular, popular latent variable model - topic models [12]. The generative model over word documents is the following. For each document in the corpus, a proportion of topics $\gamma_1, \gamma_2, \ldots, \gamma_k$ is sampled according to a prior distribution $\alpha$. Then, for each position $p$ in the document, we pick a topic $Z_p$ according to a multinomial with parameters $\gamma_1, \ldots, \gamma_k$. Conditioned on $Z_p = i$, we pick a word $j$ from a multinomial with parameters $(\beta_{i,1}, \beta_{i,2}, \ldots, \beta_{i,k})$ to put in position $p$. The matrix of values $\{\beta_{i,j}\}$ is known as the topic-word matrix.

The body of work on topic models is vast [11]. Prior theoretical work relevant in the context of this paper includes the sequence of works by [7],[4], as well as [2], [16], [17] and [10]. [7] and [4] assume that the topic-word matrix contains "anchor words". This means that each topic has a word which appears in that topic, and no other. [2] on the other hand work with a certain expansion assumption on the word-topic graph, which says that if one takes a subset S of topics, the number of words in the support of these topics should be at least $|S| + s_{max}$, where $s_{max}$ is the maximum support size of any topic. Neither paper needs any assumption on the topic priors, and can handle (almost) arbitrarily short documents.

The assumptions we make on the word-topic matrix will be related to the ones in the above works, but our documents will need to be long, so that the empirical counts of the words are close to their expected counts. Our priors will also be more structured. This is expected since we are trying to analyze an existing heuristic rather than develop a new algorithmic strategy. The case where the documents are short seems significantly more difficult. Namely, in that case there are two issues to consider. One is proving the variational approximation to the posterior distribution over topics is not too bad. The second is proving that the updates do actually reach the global optimum. Assuming long documents allows us to focus on the second issue alone, which is already challenging. On a high level, the instances we consider will have the following structure:

- The topics will satisfy a weighted expansion property: for any set S of topics of constant size, for any topic $i$ in this set, the probability mass on words which belong to $i$, and no other topic in $S$ will be large. (Similar to the expansion in [2], but only over constant sized subsets.)
- The number of topics per document will be small. Further, the probability of including a given topic in a document is almost independent of any other topics that might be included in the document already. Similar properties are satisfied by the Dirichlet prior, one of the most popular

priors in topic modeling. (Originally introduced by [12].) The documents will also have a "dominating topic", similarly as in [10].

- For each word $j$, and a topic $i$ it appears in, there will be a decent proportion of documents that contain topic $i$ and no other topic containing $j$. These can be viewed as "local anchor documents" for that word-pair topic.

We state below, informally, our main result. See Sections 6 and 7 for more details.

**Theorem.** *Under the above mentioned assumptions, popular variants of variational inference for topic models, with suitable initializations, provably recover the ground truth model in polynomial time.*

## 4 Variational relaxation for learning topic models

In this section we briefly review the variational relaxation for topic models, following closely [12]. Throughout the paper, we will denote by $N$ the total number of words and $K$ the number of topics. We will assume that we are working with a sample set of $D$ documents. We will also denote by $\tilde{f}_{d,j}$ the fractional count of word $j$ in document $d$ (i.e. $\tilde{f}_{d,j} = \text{Count}(j)/N_d$, where $\text{Count}(j)$ is the number of times word j appears in the document, and $N_d$ is the number of words in the document).

For topic models variational updates are a way to approximate the computationally intractable E-step [23] as described in Section 2. Recall the model parameters for topic models are the topic prior parameters $\alpha$ and the topic-word matrix $\beta$. The observable $X$ is the list of words in the document. The latent variables are the topic assignments $Z_j$ at each position $j$ in the document and the topic proportions $\gamma$. The variational E-step hence becomes $\tilde{P}^t(Z,\gamma) = min_{P^t \in F} KL(P^t(Z,\gamma)||P(Z,\gamma|X,\alpha^t,\beta^t)$ for some family $F$ of distributions. The family $F$ one usually considered is $P^t(\gamma,Z) = q(\gamma)\Pi_{j=1}^{N_d} q'_j(Z_j)$, i.e. a mean field family. In [12] it's shown that for Dirichlet priors $\alpha$ the optimal distributions $q, q'_j$ are a Dirichlet distribution for $q$, with some parameter $\tilde{\gamma}$, and multinomials for $q'_j$, with some parameters $\phi_j$. The variational EM updates are shown to have the following form. In the E-step, one runs to convergence the following updates on the $\phi$ and $\tilde{\gamma}$ parameters: $\phi_{d,j,i} \propto \beta_{i,w_{d,j}}^t e^{E_q[\log(\gamma_d)|\tilde{\gamma}_d]}$, $\gamma_{\tilde{d},i} = \alpha_{d,i}^t + \sum_{j=1}^{N_d} \phi_{d,j,i}$. In the M-step, one

updates the $\beta$ and parameters by setting $\beta_{i,j}^{t+1} \propto \sum_{d=1}^{D} \sum_{j'=1}^{N_d} \phi_{d,j,i}^t w_{d,j,j'}$ where $\phi_{d,j,i}^t$ is the converged

value of $\phi_{d,j,i}$; $w_{d,j}$ is the word in document $d$, position $j$; $w_{d,j,j'}$ is an indicator variable which is 1 if the word in position $j'$ in document $d$ is word $j$. The $\alpha$ Dirichlet parameters do not have a closed form expression and are updated via gradient descent.

### 4.1 Simplified updates in the long document limit

From the above updates it is difficult to give assign an intuitive meaning to the $\tilde{\gamma}$ and $\phi$ parameters. (Indeed, it's not even clear what one would like them to be ideally at the global optimum.) We will be however working in the large document limit - and this will simplify the updates. In particular, in the E-step, in the large document limit, the first term in the update equation for $\tilde{\gamma}$ has a vanishing contribution. In this case, we can simplify the E-update as: $\phi_{d,j,i} \propto \beta_{i,j}^t \gamma_{d,i}$, $\gamma_{d,i} \propto \sum_{j=1}^{N_d} \phi_{d,j,i}$. Notice, importantly, in the second update we now use variables $\gamma_{d,i}$ instead of $\tilde{\gamma}_{d,i}$, which are normalized such that $\sum_{i=1}^{K} \gamma_{d,i} = 1$. These correspond to the max-likelihood topic proportions, given our current estimates $\beta_{i,j}^t$ for the model parameters. The M-step will remain as is - but we will focus on the $\beta$ only, and ignore the $\alpha$ updates - as the $\alpha$ estimates disappeared from the E updates: $\beta_{i,j}^{t+1} \propto \sum_{d=1}^{D} \tilde{f}_{d,j} \gamma_{d,i}^t$, where $\gamma_{d,i}^t$ is the converged value of $\gamma_{d,i}$. In this case, the intuitive meaning of the $\beta^t$ and $\gamma^t$ variables is clear: they are estimates of the the model parameters, and the max-likelihood topic proportions, given an estimate of the model parameters, respectively.

The way we derived them, these updates appear to be an approximate form of the variational updates in [12]. However it is possible to also view them in a more principled manner. These updates

approximate the posterior distribution $P(Z, \gamma | X, \alpha^t, \beta^t)$ by first approximating this posterior by $P(Z | X, \gamma^*, \alpha^t, \beta^t)$, where $\gamma^*$ is the max-likelihood value for $\gamma$, given our current estimates of $\alpha, \beta$, and then setting $P(Z | X, \gamma^*, \alpha^t, \beta^t)$ to be a product distribution. It is intuitively clear that in the large document limit, this approximation should not be much worse than the one in [12], as the posterior concentrates around the maximum likelihood value. (And in fact, our proofs will work for finite, but long documents.) Finally, we will rewrite the above equations in a slightly more convenient form. Denoting $f_{d,j} = \sum_{i=1}^{K} \gamma_{d,i} \beta_{i,j}^t$, the E-step can be written as: iterate until convergence $\gamma_{d,i} = \gamma_{d,i} \sum_{j=1}^{N} \frac{\tilde{f}_{d,j}}{f_{d,j}} \beta_{i,j}^t$. The M-step becomes: $\beta_{i,j}^{t+1} = \beta_{i,j}^t \frac{\sum_{d=1}^{D} \frac{\tilde{f}_{d,j}}{f_{d,j}^t} \gamma_{d,i}^t}{\sum_{d=1}^{D} \gamma_{d,i}^t}$ where $f_{d,j}^t = \sum_{i=1}^{K} \gamma_{d,i}^t \beta_{i,j}^t$ and $\gamma_{d,i}^t$ is the converged value of $\gamma_{d,i}$.

## 4.2 Alternating KL minimization and thresholded updates

We will further modify the E and M-step update equations we derived above. In a slightly modified form, these updates were used in a paper by [21] in the context of non-negative matrix factorization. There the authors proved that under these updates $\sum_{d=1}^{D} KL(f_{d,j}^t || \tilde{f}_{d,j})$ is non-decreasing. One can easily modify their arguments to show that the same property is preserved if the E-step is replaced by a step $\gamma_d^t = \min_{\gamma_d^t \in \Delta_K} KL(\tilde{f}_d || f_d)$, where $\Delta_K$ is the K-dimensional simplex - i.e. minimizing the KL divergence between the counts and the "predicted counts" with respect to the $\gamma$ variables. (In fact, iterating the $\gamma$ updates above is a way to solve this convex minimization problem via a version of gradient descent which makes multiplicative updates, rather than additive updates.)

Thus the updates are performing alternating minimization using the KL divergence as the distance measure (with the difference that for the $\beta$ variables one essentially just performs a single gradient step). In this paper, we will make a modification of the M-step which is very natural. Intuitively, the update for $\beta_{i,j}^t$ goes over all appearances of the word $j$ and adds the "fractional assignment" of the word $j$ to topic $i$ under our current estimates of the variables $\beta, \gamma$. In the modified version we will only average over those documents $d$, where $\gamma_{d,i}^t > \gamma_{d,i'}^t, \forall i' \neq i$. The intuitive reason behind this modification is the following. The EM updates we are studying work with the KL divergence, which puts more weight on the larger entries. Thus, for the documents in $D_i$, the estimates for $\gamma_{d,i}^t$ should be better than they might be in the documents $D \setminus D_i$. (Of course, since the terms $f_{d,j}^t$ involve all the variables $\gamma_{d,i}^t$, it is not a priori clear that this modification will gain us much, but we will prove that it in fact does.) Formally, we discuss the three modifications of variational inference specified as Algorithm 1, 2 and 3 (we call them tEM, for thresholded EM):

---

**Algorithm 1** KL-tEM

---

(E-step) Solve the following convex program for each document $d$: $\min_{\gamma_{d,i}^t} \sum_j \tilde{f}_{d,j} \log(\frac{\tilde{f}_{d,j}}{f_{d,j}^t})$, s.t. $\gamma_{d,i}^t \geq 0, \sum_i \gamma_{d,i}^t = 1$ and $\gamma_{d,i}^t = 0$ if $i$ is not in the support of document $d$

(M-step) Let $D_i$ to be the set of documents $d$, s.t. $\gamma_{d,i}^t > \gamma_{d,i'}^t, \forall i' \neq i$.

Set $\beta_{i,j}^{t+1} = \beta_{i,j}^t \frac{\sum_{d \in D_i} \frac{\tilde{f}_{d,j}}{f_{d,j}^t} \gamma_{d,i}^t}{\sum_{d \in D_i} \gamma_{d,i}^t}$

---

## 5 Initializations

We will consider two different strategies for initialization. First, we will consider the case where we initialize with the topic-word matrix, and the document priors having the correct support. The analysis of tEM in this case will be the cleanest. While the main focus of the paper is tEM, we'll show that this initialization can actually be done for our case efficiently. Second, we will consider an initialization that is inspired by what the current LDA-c implementation uses. Concretely, we'll

---
**Algorithm 2** Iterative tEM
---
(E-step) Initialize $\gamma_{d,i}$ uniformly among the topics in the support of document $d$.
Repeat

$$\gamma_{d,i} = \gamma_{d,i} \sum_{j=1}^{N} \frac{\tilde{f}_{d,j}}{f_{d,j}} \beta_{i,j}^{t} \tag{4.1}$$

until convergence.
(M-step) Same as above.

---
**Algorithm 3** Incomplete tEM
---
(E-step) Initialize $\gamma_{d,i}$ with the values gotten in the previous iteration, then perform just one step of 4.1.
(M-step) Same as before.

---

assume that the user has some way of finding, for each topic $i$, a *seed document* in which the proportion of topic $i$ is at least $C_l$. Then, when initializing, one treats this document as if it were pure: namely one sets $\beta_{i,j}^0$ to be the fractional count of word $j$ in this document. We do not attempt to design an algorithm to find these documents.

## 6 Case study 1: Sparse topic priors, support initialization

We start with a simple case. As mentioned, all of our results only hold in the long documents regime: we will assume for each document $d$, the number of sampled words is large enough, so that one can approximate the expected frequencies of the words, i.e., one can find values $\gamma_{d,i}^*$, such that $\tilde{f}_{d,j} = (1 \pm \epsilon) \sum_{i=1}^{K} \gamma_{d,i}^* \beta_{i,j}^*$. We'll split the rest of the assumptions into those that apply to the topic-word matrix, and the topic priors. Let's first consider the assumptions on the topic-word matrix. We will impose conditions that ensure the topics don't overlap too much. Namely, we assume:

- *Words are discriminative*: Each word appears in $o(K)$ topics.
- *Almost disjoint supports*: $\forall i, i'$, if the intersection of the supports of $i$ and $i'$ is S, $\sum_{j \in S} \beta_{i,j}^* \leq o(1) \cdot \sum_j \beta_{i,j}^*$.

We also need assumptions on the topic priors. The documents will be sparse, and all topics will be roughly equally likely to appear. There will be virtually no dependence between the topics: conditioning on the size or presence of a certain topic will not influence much the probability of another topic being included. These are analogues of distributions that have been analyzed for dictionary learning [6]. Formally:

- *Sparse and gapped documents*: Each of the documents in our samples has at most $T = O(1)$ topics. Furthermore, for each document $d$, the largest topic $i_0 = \text{argmax}_i \gamma_{d,i}^*$ is such that for any other topic $i'$, $\gamma_{d,i'}^* - \gamma_{d,i_0}^* > \rho$ for some (arbitrarily small) constant $\rho$.
- *Dominant topic equidistribution*: The probability that topic $i$ is such that $\gamma_{d,i}^* > \gamma_{d,i'}^*, \forall i' \neq i$ is $\Theta(1/K)$.
- *Weak topic correlations and independent topic distribution*: For all sets S with $o(K)$ topics, it must be the case that: $\mathbf{E}[\gamma_{d,i}^* | \gamma_{d,i}^* \text{ is dominating}] = (1 \pm o(1))\mathbf{E}[\gamma_{d,i}^* | \gamma_{d,i}^* \text{ is dominating}, \gamma_{d,i'}^* = 0, i' \in S]$. Furthermore, for any set S of topics, s.t. $|S| \leq T - 1$, $\Pr[\gamma_{d,i}^* > 0 | \gamma_{d,i'}^* \forall i' \in S] = \Theta(\frac{1}{K})$

These assumptions are a less smooth version of properties of the Dirichlet prior. Namely, it's a folklore result that Dirichlet draws are sparse with high probability, for a certain reasonable range of parameters. This was formally proven by [25] - though sparsity there means a small number of large coordinates. It's also well known that Dirichlet essentially cannot enforce any correlation between different topics. [1]

The above assumptions can be viewed as a *local* notion of separability of the model, in the following sense. First, consider a particular document $d$. For each topic $i$ that participates in that document, consider the words $j$, which only appear in the support of topic $i$ in the document. In some sense, these words are *local anchor words* for that document: these words appear only in one topic of that document. Because of the "almost disjoint supports" property, there will be a decent mass on these words in each document. Similarly, consider a particular non-zero element $\beta_{i,j}^*$ of the topic-word matrix. Let's call $D_l$ the set of documents where $\beta_{i',j}^* = 0$ for all other topics $i' \neq i$ appearing in that document. These documents are like *local anchor documents* for that word-topic pair: in those documents, the word appears as part of only topic $i$. It turns out the above properties imply there is a decent number of these for any word-topic pair.

Finally, a technical condition: we will also assume that all nonzero $\gamma_{d,i}^*, \beta_{i,j}^*$ are at least $\frac{1}{poly(N)}$. Intuitively, this means if a topic is present, it needs to be reasonably large, and similarly for words in topics. Such assumptions also appear in the context of dictionary learning [6].

We will prove the following

**Theorem 1.** *Given an instance of topic modelling satisfying the properties specified above, where the number of documents is $\Omega(\frac{K \log^2 N}{\epsilon^2})$, if we initialize the supports of the $\beta_{i,j}^t$ and $\gamma_{d,i}^t$ variables correctly, after $O\left(\log(1/\epsilon') + \log N\right)$ KL-tEM, iterative-tEM updates or incomplete-tEM updates, we recover the topic-word matrix and topic proportions to multiplicative accuracy $1 + \epsilon'$, for any $\epsilon'$ s.t. $1 + \epsilon' \leq \frac{1}{(1-\epsilon)^7}$.*

**Theorem 2.** *If the number of documents is $\Omega(K^4 \log^2 K)$, there is a polynomial-time procedure which with probability $1 - \Omega(\frac{1}{K})$ correctly identifies the supports of the $\beta_{i,j}^*$ and $\gamma_{d,i}^*$ variables.*

**Provable convergence of tEM:** The correctness of the tEM updates is proven in 3 steps:

- *Identifying dominating topic*: First, we prove that if $\gamma_{d,i}^t$ is the largest one among all topics in the document, topic $i$ is actually the largest topic.
- *Phase I: Getting constant multiplicative factor estimates*: After initialization, after $O(\log N)$ rounds, we will get to variables $\beta_{i,j}^t, \gamma_{d,i}^t$ which are within a constant multiplicative factor from $\beta_{i,j}^*, \gamma_{d,i}^*$.
- *Phase II (Alternating minimization - lower and upper bound evolution)*: Once the $\beta$ and $\gamma$ estimates are within a constant factor of their true values, we show that the lone words and documents have a *boosting* effect: they cause the multiplicative upper and lower bounds to improve at each round.

The updates we are studying are multiplicative, not additive in nature, and the objective they are optimizing is non-convex, so the standard techniques do not work. The intuition behind our proof in Phase II can be described as follows. Consider one update for one of the variables, say $\beta_{i,j}^t$. We show that $\beta_{i,j}^{t+1} \approx \alpha\beta_{i,j}^* + (1-\alpha)C^t\beta_{i,j}^*$ for some constant $C^t$ at time step $t$. $\alpha$ is something fairly large (one should think of it as $1 - o(1)$), and comes from the existence of the local anchor documents. A similar equation holds for the $\gamma$ variables, in which case the "good" term comes from the local anchor words. Furthermore, we show that the error in the $\approx$ decreases over time, as does the value of $C^t$, so that eventually we can reach $\beta_{i,j}^*$. The analysis bears a resemblance to the *state evolution* and *density evolution* methods in error decoding algorithm analysis - in the sense that we maintain a quantity about the evolving system, and analyze how it evolves under the specified iterations. The quantities we maintain are quite simple - upper and lower multiplicative bounds on our estimates at any round $t$.

**Initialization:** Recall the goal of this phase is to recover the supports - i.e. to find out which topics are present in a document, and identify the support of each topic. We will find the topic supports first. This uses an idea inspired by [8] in the setting of dictionary learning. Roughly, we devise a test, which will take as input two documents $d, d'$, and will try to determine if the two documents have a topic in common or not. The test will have no false positives, i.e., will never say YES, if the documents don't have a topic in common, but might say NO even if they do. We then ensure that with high probability, for each topic we find a pair of documents intersecting in that topic, such that the test says YES. [2]

# 7 Case study 2: Dominating topics, seeded initialization

Next, we'll consider an initialization which is essentially what the current implementation of LDA-c uses. Namely, we will call the following initialization a *seeded* initialization:

- For each topic $i$, the user supplies a document $d$, in which $\gamma_{d,i}^* \geq C_l$.
- We treat the document as if it only contains topic $i$ and initialize with $\beta_{i,j}^0 = f_{d,j}^*$.

We show how to modify the previous analysis to show that with a few more assumptions, this strategy works as well. Firstly, we will have to assume anchor words, that make up a decent fraction of the mass of each topic. Second, we also assume that the words have a bounded *dynamic range*, i.e. the values of a word in two different topics are within a constant $B$ from each other. The documents are still gapped, but the gap now must be larger. Finally, in roughly $1/B$ fraction of the documents where topic $i$ is dominant, that topic has proportion $1 - \delta$, for some small (but still constant) $\delta$. A similar assumption (a small fraction of almost pure documents) appeared in a recent paper by [10]. Formally, we have:

- *Small dynamic range and large fraction of anchors*: For each discriminative words, if $\beta_{i,j}^* \neq 0$ and $\beta_{i',j}^* \neq 0$, $\beta_{i,j}^* \leq B\beta_{i',j}^*$. Furthermore, each topic $i$ has anchor words, such that their total weight is at least $p$.
- *Gapped documents*: In each document, the largest topic has proportion at least $C_l$, and all the other topics are at most $C_s$, s.t.

$$C_l - C_s \geq \frac{1}{p}\left(\sqrt{2\left(p\log(\frac{1}{C_l}) + (1-p)\log(BC_l)\right)} + \sqrt{\log(1+\epsilon)}\right) + \epsilon$$

- *Small fraction of $1 - \delta$ dominant documents*: Among all the documents where topic $i$ is dominating, in a $8/B$ fraction of them, $\gamma_{d,i}^* \geq 1 - \delta$, where

$$\delta := \min\left(\frac{C_l^2}{2B^3} - \frac{1}{p}\left(\sqrt{2\left(p\log(\frac{1}{C_l}) + (1-p)\log(BC_l)\right)} + \sqrt{\log(1+\epsilon)}\right) - \epsilon, 1 - \sqrt{C_l}\right)$$

The dependency between the parameters $B, p, C_l$ is a little difficult to parse, but if one thinks of $C_l$ as $1-\eta$ for $\eta$ small, and $p \geq 1 - \frac{\eta}{\log B}$, since $\log(\frac{1}{C_l}) \approx 1+\eta$, roughly we want that $C_l - C_s \gg \frac{2}{p}\sqrt{\eta}$. (In other words, the weight we require to have on the anchors depends only *logarithmically* on the range $B$.) In the documents where the dominant topic has proportion $1 - \delta$, a similar reasoning as above gives that we want is approximately $\gamma_{d,i}^* \geq 1 - \frac{1-2\eta}{2B^3} + \frac{2}{p}\sqrt{\eta}$. The precise statement is as follows:

**Theorem 3.** *Given an instance of topic modelling satisfying the properties specified above, where the number of documents is $\Omega(\frac{K\log^2 N}{\epsilon^2})$, if we initialize with seeded initialization, after $O\left(\log(1/\epsilon') + \log N\right)$ of KL-tEM updates, we recover the topic-word matrix and topic proportions to multiplicative accuracy $1 + \epsilon'$, if $1 + \epsilon' \geq \frac{1}{(1-\epsilon)^7}$.*

The proof is carried out in a few phases:

- *Phase I: Anchor identification*: We show that as long as we can identify the dominating topic in each of the documents, anchor words will make progress: after $O(\log N)$ number of rounds, the values for the topic-word estimates will be almost zero for the topics for which word $w$ is not an anchor. For topic for which a word is an anchor we'll have a good estimate.
- *Phase II: Discriminative word identification*: After the anchor words are properly identified in the previous phase, if $\beta_{i,j}^* = 0$, $\beta_{i,j}^t$ will keep dropping and quickly reach almost zero. The values corresponding to $\beta_{i,j}^* \neq 0$ will be decently estimated.
- *Phase III: Alternating minimization*: After Phase I and II above, we are back to the scenario of the previous section: namely, there is improvement in each next round.

During Phase I and II the intuition is the following: due to our initialization, even in the beginning, each topic is "correlated" with the correct values. In a $\gamma$ update, we are minimizing $KL(\tilde{f}_d||f_d)$ with respect to the $\gamma_d$ variables, so we need a way to argue that whenever the $\beta$ estimates are not too bad, minimizing this quantity provides an estimate about how far the optimal $\gamma_d$ variables are from $\gamma_d^*$. We show the following useful claim:

**Lemma 4.** *If, for all topics $i$, $KL(\beta_i^*||\beta_i^t) \leq R_\beta$, and $\min_{\gamma_d \in \Delta_K} KL(\tilde{f}_{d,j}||f_{d,j}) \leq R_f$, after running a KL divergence minimization step with respect to the $\gamma_d$ variables, we get that $||\gamma_d^* - \gamma_d||_1 \leq \frac{1}{p}(\sqrt{\frac{1}{2}R_\beta} + \frac{1}{2}\sqrt{R_f}) + \epsilon$.*

This lemma critically uses the existence of anchor words - namely we show $||\beta^* v||_1 \geq p||v||_1$. Intuitively, if one thinks of $v$ as $\gamma^* - \gamma^t$, $||\beta^* v||_1$ will be large if $||v||_1$ is large. Hence, if $||\beta^* - \beta^t||_1$ is not too large, whenever $||f^* - f^t||_1$ is small, so is $||\gamma^* - \gamma^t||_1$. We will be able to maintain $R_\beta$ and $R_f$ small enough throughout the iterations, so that we can identify the largest topic in each of the documents.

## 8   On common words

We briefly remark on *common words*: words such that $\beta_{i,j}^* \leq \kappa \beta_{i',j}^*, \forall i, i', \kappa \leq B$. In this case, the proofs above, as they are, will not work, [3] since common words do not have any lone documents. However, if $1 - \frac{1}{\kappa^{100}}$ fraction of the documents where topic $i$ is dominant contains topic $i$ with proportion $1 - \frac{1}{\kappa^{100}}$ and furthermore, in each topic, the weight on these words is no more than $\frac{1}{\kappa^{100}}$, then our proofs still work with either initialization[4] The idea for the argument is simple: when the dominating topic is very large, we show that $\frac{f_{d,j}^*}{f_{d,j}^t}$ is very highly correlated with $\frac{\beta_{i,j}^*}{\beta_{i,j}^t}$, so these documents behave like anchor documents. Namely, one can show:

**Theorem 5.** *If we additionally have common words satisfying the properties specified above, after $O(\log(1/\epsilon') + \log N)$ KL-tEM updates in Case Study 2, or any of the tEM variants in Case Study 1, and we use the same initializations as before, we recover the topic-word matrix and topic proportions to multiplicative accuracy $1 + \epsilon'$, if $1 + \epsilon' \geq \frac{1}{(1-\epsilon)^7}$.*

## 9   Discussion and open problems

In this work we provide the first characterization of sufficient conditions when variational inference leads to optimal parameter estimates for topic models. Our proofs also suggest possible hard cases for variational inference, namely instances with large dynamic range compared to the proportion of anchor words and/or correlated topic priors. It's not hard to hand-craft such instances where support initialization performs very badly, even with only anchor and common words. We made no effort to explore the optimal relationship between the dynamic range and the proportion of anchor words, as it's not clear what are the "worst case" instances for this trade-off.

Seeded initialization, on the other hand, empirically works much better. We found that when $C_l \geq 0.6$, and when the proportion of anchor words is as low as $0.2$, variational inference recovers the ground truth, even on instances with fairly large dynamic range. Our current proof methods are too weak to capture this observation. (In fact, even the largest topic is sometimes misidentified in the initial stages, so one cannot even run tEM, only the vanilla variational inference updates.) Analyzing the dynamics of variational inference in this regime seems like a challenging problem which would require significantly new ideas.

## Footnotes

[1] We show analogues of the weak topic correlations property and equidistribution in the supplementary material for completeness sake.

[2]The detailed initialization algorithm is included in the supplementary material.

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
