[Supplementary Material]

# Supplementary material

# Contents

# 1   Notation throughout supplementary material

We will use $\simeq, \lesssim, \gtrsim$ to denote that the corresponding (in)equality holds up to constants. We will use $\Leftrightarrow$ to denote equivalence. We will say that an event happens *with high probability*, if it happens with probability $1 - \frac{1}{K^c}$ or $1 - \frac{1}{N^c}$ for some constant $c$.

# 2   Case study 1: Sparse topic priors, support initialization

## 2.1   Provable convergence of tEM

As a reminder, the theorem we want to prove is:

**Theorem 1.** *Given an instance of topic modelling satisfying the Case Study 1 properties specified above, where the number of documents is $\Omega(\frac{K \log^2 N}{\epsilon'^2})$, if we initialize the supports of the $\beta_{i,j}^t$ and $\gamma_{d,i}^t$ variables correctly, after $O(\log(1/\epsilon') + \log N)$ KL-tEM, iterative-tEM updates or incomplete-tEM updates, we recover the topic-word matrix and topic proportions to multiplicative accuracy $1 + \epsilon'$, for any $\epsilon'$ s.t. $1 + \epsilon' \le \frac{1}{(1-\epsilon)^7}$.*

The general outline of the proof will be the following.

- *Identifying dominating topic*: For the modified tEM updates, we need to make sure that the topic with maximal $\gamma_{d,i}^t$ is the dominant.

- *Phase I: Getting constant multiplicative factor estimates*: First, we'll show that after initialization, after $O(\log N)$ number of rounds, we will get to variables $\beta_{i,j}^t$, $\gamma_{d,i}^t$ which are within a constant multiplicative factor from $\beta_{i,j}^*$, $\gamma_{d,i}^*$.

  - *Lower bounds on the $\beta$ and $\gamma$ variables*: We'll show that determining the supports of the documents and the topic-word matrix, as well as being able to identify the documents in which topic $i$ is large is enough to ensure that all the $\beta_{i,j}^t$ and $\gamma_{i,j}^t$ variables are lower bounded by $\frac{1}{C_\beta^0} \beta_{i,j}^*$ and $\frac{1}{C_\gamma^0} \gamma_{d,i}^*$ respectively for some constants $C_\beta^0 \ge 1, C_\gamma^0 \ge 1$.

  - *Improving upper bounds on the $\beta_{i,j}^t$ values*: We show that, if the above two properties are satisfied, we can get a multiplicative upper bound of the $\beta_{i,j}^t$ values, which strictly improves at each step until it reaches a constant. This improvement is very fast: we only need a logarithmic number of steps. After this happens, we show that the $\gamma$ variables corresponding to these $\beta$ estimates must be within a constant of the ground truth as well.

- *Phase II (Alternating minimization - lower and upper bound evolution)*: Once the $\beta$ and $\gamma$ estimates are within a constant factor of their true values, we show that the lone words and documents have a *boosting* effect: they cause the multiplicative upper and lower bounds to improve at each round.

A word about incorporating the "correct supports" assumption in our algorithms. For the $\beta$ variables this is obvious: we just set $\beta_{i,j}^t = 0$ if $\beta_{i,j}^* = 0$. For the $\gamma$ variables it's also fairly straightforward. In KL-tEM we mean simply that in the convex program above, we constrain $\gamma_{d,i}^t = 0$ if $\gamma_{d,i}^* = 0$.

In the iterative version, this just means that before starting the $\gamma$ iterations, we set the initial value to 0 if $\gamma_{d,i}^* = 0$, and uniform among the rest of the variables. Same for the incomplete version.

In the interest of brevity, whenever we say "the supports are correct", the above is what we will mean.

Recall, we use $t$ to count the iterations for $\beta$ variables. Put another way, $\gamma_{d,i}^t$ is the value we get for $\gamma_{d,i}$ after the $\beta$ variables were updated to $\beta_{i,j}^t$. (Which of course, implies, $\beta_{i,j}^{t+1}$ will be the values we get for the $\beta$ variables after the $\gamma$ variables are updated to $\gamma_{d,i}^t$.)

The proofs are for each of the variants of tEM are similar. For starters, we show everything for KL-tEM, and then just mention how to modify the arguments to get the results for the other variants in section 2.2.

### 2.1.1 Determining largest topic

First, we show that the "thresholding" operation works. Namely, we show that if $\gamma_{d,i}^t > \gamma_{d,i'}^t, \forall i \neq i'$, then $\gamma_{d,i}^*$ is the largest topic in the document (there is a unique one by the "slightly gapped documents" property). Furthermore, we can say that $\frac{1}{2}\gamma_{d,i}^* \leq \gamma_{d,i}^t \leq 2\gamma_{d,i}^*$.

**Lemma 2.** *Fix a document $d$. Let the supports of the $\gamma$ and $\beta$ variables be correct. Then, after a $\gamma$ iteration, if $\gamma_{d,i}^t > \gamma_{d,i'}^t, i \neq i'$, $\gamma_{d,i}^*$ is the largest topic in the document. Furthermore, $\frac{1}{2}\gamma_{d,i}^* \leq \gamma_{d,i}^t \leq 2\gamma_{d,i}^*$.*

*Proof.* Since there are a constant number of topics in the document, the largest topic has proportion $\Omega(1)$.

Consider the KL-tEM convex optimization problem. The KKT conditions are easily seen to imply[1]:

$$\sum_{j=1}^N \frac{\tilde{f}_{d,j}}{f_{d,j}^t}\beta_{i,j}^t = 1 \qquad (2.1)$$

For each topic $i$, since we are considering a constrained optimization problem, it has to be the case that it either satisfies 2.1, $\gamma_{d,i}^t = 0$ or $\gamma_{d,i}^t = 1$.

Let's assume first that $i$ satisfies 2.1. Then,

$$\gamma_{d,i}^t = \sum_{j=1}^N \frac{\tilde{f}_{d,j}}{f_{d,j}^t}\beta_{i,j}^t\gamma_{d,i}^t \leq \sum_{j:\beta_{i,j}^* \neq 0} \tilde{f}_{d,j}$$

Let's call the words $j$, which only appear in the support of topic $i$ in the document *lone* for that topic, and let's denote that set as $L_i$.

If $L_i$ are the lone words for topic $i$, $\sum_{j \notin L_i, \beta_{i,j}^* \neq 0} \tilde{f}_{d,j} = To(1) = o(1)$, so

$$\gamma_{d,i}^t \leq \sum_{j \in L_i}(1+\epsilon)\beta_{i,j}^*\gamma_{d,i}^* + o(1) \leq (1+\epsilon)\gamma_{d,i}^* + o(1) \leq \gamma_{d,i}^* + o(1)$$

On the other hand, $\gamma_{d,i}^t \geq \sum_{j \in L_i}\beta_{i,j}^*\gamma_{d,i}^* \geq (1-\epsilon)(1-o(1))\gamma_{d,i}^* \geq (1-o(1))\gamma_{d,i}^*$, so $\gamma_{d,i}^t \geq \gamma_{d,i}^* - o(1)$.

Since there is a constant gap of $\rho$ between the largest topic and the next largest one, the maximum $\gamma_{d,i}^t$ is indeed the largest topic in the document. Furthermore, since $(1-o(1))\gamma_{d,i}^* \leq \gamma_{d,i}^t \leq (1+o(1))\gamma_{d,i}^*$, clearly $\frac{1}{2}\gamma_{d,i}^* \leq \gamma_{d,i}^t \leq 2\gamma_{d,i}^*$ follows as well.

On the other hand, we claim no topic which is in the support of a document $d$ can actually have $\gamma_{d,i}^t = 0$.

If this happens, it's easy to see that $\sum_{j=1}^N \tilde{f}_{d,j}\log(\frac{\tilde{f}_{d,j}}{f_{d,j}^t}) = \infty$: one only needs to look at a summand corresponding to a lone word $j$ for topic $i$. Just by virtue of the way lone words are defined, $\gamma_{d,i}^t = 0$ would imply $f_{d,j}^t = 0$. It's clear that one can get a finite value for $\sum_j \tilde{f}_{d,j}\log(\frac{\tilde{f}_{d,j}}{f_{d,j}^t})$ on the other hand, by just setting $\gamma_{d,i}^t = \gamma_{d,i}^*$, so $\gamma_{d,i}^t = 0$ cannot happen at an optimum. $\qquad\square$

### 2.1.2 Lower bounds on the $\gamma_{d,i}^t$ and $\beta_{i,j}^t$ variables

Next, we show that subject to the thresholding being correct, at any point in time $t$, all the estimates $\gamma_{d,i}^t$ and $\beta_{i,j}^t$ are appropriately lower bounded.

The proof is similar for both the $\beta$ and $\gamma$ variables, and both for the KL-tEM and iterative tEM updates, but as mentioned before, we focus on the KL-tEM first.

**Lemma 3.** *Fix a particular document $d$. Suppose that the supports of the $\gamma$ and $\beta$ variables are correct. Then, $\gamma_{d,i}^t \geq (1-o(1))\gamma_{d,i}^*$.*

*Proof.* Multiplying both sides of 2.1 by $\gamma_{d,i}^t$, we get

$$\gamma_{d,i}^t = \sum_{j=1}^N \frac{\tilde{f}_{d,j}}{f_{d,j}^t} \beta_{i,j}^t \gamma_{d,i}^t$$

As above, let's split the above sum in two parts: lone words, and non-lone. Then clearly,

$$\gamma_{d,i}^t \geq \sum_{j \in L_i} (1 - \epsilon) \beta_{i,j}^* \gamma_{d,i}^*$$

For notational convenience, let's denote $\tilde{\alpha} = \sum_{j \in L_i} \beta_{i,j}^*$. Let's estimate $\tilde{\alpha}$. By the assumption on the size of the intersection of topics,

$$\sum_{j \notin L_i} \beta_{i,j}^* \leq Tr = o(1)$$

.

Hence, $\tilde{\alpha} \geq (1 - \epsilon)(1 - o(1)) = 1 - o(1)$. So, the claim of the lemma holds.

$\square$

The lower bound on the $\beta_{i,j}^t$ values proceeds similarly, but here we will crucially make use of the fact that for the large topics, we have both upper and lower bounds on the $\gamma_{d,i}^t$ values.

**Lemma 4.** *Suppose that the supports of the $\gamma$ and $\beta$ variables are correct. Additionally, if $i$ is a large topic in $d$, let $\frac{1}{2}\gamma_{d,i}^* \leq \gamma_{d,i}^t \leq 2\gamma_{d,i}^*$. Then, $\beta_{i,j}^{t+1} \geq \frac{1}{2}(1 - o(1))\beta_{i,j}^*$.*

*Proof.* Let's call *lone* the documents where $\beta_{i',j}^* = 0$ for all other topics $i' \neq i$ appearing in that document for the topic-word pair $(i, j)$. Let $D_l$ be the set of lone documents. Then, certainly it's true that

$$\beta_{i,j}^{t+1} \geq \beta_{i,j}^t \frac{\sum_{d \in D_l} \frac{\tilde{f}_{d,j}}{f_{d,j}^t} \gamma_{d,i}^t}{\sum_{d=1}^D \gamma_{d,i}^t}$$

However, for a lone document, $f_{d,j}^t = \gamma_{d,i}^t \cdot \beta_{i,j}^t$ (it's easy to check all the other terms in the summation for $f_{d,j}^t$ vanish, because either $\gamma_{d,i'}^t = 0$ or $\beta_{i',j}^t = 0$). Hence,

$$\beta_{i,j}^{t+1} \geq \frac{\sum_{d \in D_l} (1 - \epsilon) \frac{\gamma_{d,i}^* \beta_{i,j}^*}{\gamma_{d,i}^t \cdot \beta_{i,j}^t} \beta_{i,j}^t \gamma_{d,i}^t}{\sum_{d=1}^D \gamma_{d,i}^t} = (1 - \epsilon)\beta_{i,j}^* \frac{\sum_{d \in D_l} \gamma_{d,i}^*}{\sum_{d=1}^D \gamma_{d,i}^t}$$

However, since the update is happening only over documents where topic $i$ is large, $\gamma_{d,i}^t \leq 2\gamma_{d,i}^*$. So, we can conclude

$$\beta_{i,j}^{t+1} \geq (1 - \epsilon)\beta_{i,j}^* \frac{1}{2} \frac{\sum_{d \in D_l} \gamma_{d,i}^*}{\sum_{d=1}^D \gamma_{d,i}^*}$$

Let's call $\alpha = \frac{\sum_{d \in D_l} \gamma_{d,i}^*}{\sum_{d=1}^D \gamma_{d,i}^*}$, and let's analyze it's value.
By Lemma 49 and Lemma 48,

$$\sum_{d \in D_l} \gamma_{d,i}^* \geq (1 - \epsilon)|D_l|\mathbf{E}[\gamma_{d,i}^*|\gamma_{d,i}^* \text{ is dominating}, \gamma_{d,i'}^* = 0, \forall i' \neq i \text{ s.t. } j \text{ appears in topic } i']$$

$$\sum_{d=1}^D \gamma_{d,i}^* \leq (1 + \epsilon)|D|\mathbf{E}[\gamma_{d,i}^*|\gamma_{d,i}^* \text{ is dominating}]$$

By the weak topic correlations assumption, then, $\frac{\sum_{d \in D_l} \gamma_{d,i}^*}{\sum_{d=1}^D \gamma_{d,i}^*} \geq (1 - o(1))\frac{|D_l|}{|D|}$.

Furthermore, by the independent topic inclusion property, each of the $o(K)$ topics other than $i$ that word $j$ belongs to appears in a document with probability $\Theta(1/K)$, so the probability that a document which

contains topic $i$ contains one of them is $o(1)$, i.e. $\frac{|D_l|}{|D|}$. By Lemma 50, furthermore, $\frac{|D_l|}{|D|} \geq 1 - o(1)$ when $\epsilon = o(1)$. Hence, $\alpha \geq 1 - o(1)$.

Altogether, we get that $\beta_{i,j}^{t+1} \geq \frac{1}{2}(1 - o(1))\beta_{i,j}^*$ as claimed. $\qquad \square$

### 2.1.3 Upper bound on the $\beta_{i,j}^t$ values

Having established a lower bound on the $\beta_{i,j}^t$ variables throughout all iterations, together with the lower bounds on the $\gamma_{d,i}^t$ variables and the good estimates for the large topics, we will be able to prove the upper bound of the multiplicative error of $\beta_{i,j}^t$ keeps improving, until $\beta_{i,j}^t \leq C_\beta \beta_{i,j}^*$, for some constant $C_\beta$.

**Lemma 5.** *Let the $\beta$ variables have the correct support, and $\beta_{i,j}^t \geq \frac{1}{C_m}\beta_{i,j}^*$, $\gamma_{d,i}^t \geq \frac{1}{C_m}\gamma_{d,i}^*$ whenever $\beta_{i,j}^* \neq 0$, $\gamma_{d,i}^* \neq 0$. Let $\beta_{i,j}^t = C_\beta^t \beta_{i,j}^*$, where $C_\beta^t \geq 4C_m$, and $C_m$ is a constant. Then, in the next iteration, $\beta_{i,j}^{t+1} \leq C_\beta^{t+1} \beta_{i,j}^*$, where $C_\beta^{t+1} \leq \frac{C_\beta^t}{2}$.*

*Proof.* Without loss of generality, let's assume $C_m \geq 2$. (Since certainly, if the statement of the lemma holds with a smaller constant, it holds with $C_m = 2$.)

We proceed similarly as in the prior analyses. We will split the sum into the portion corresponding to the lone and non-lone documents.

Let's analyze the terms $\frac{\tilde{f}_{d,i}}{f_{d,j}^t}\gamma_{d,i}^t$ corresponding to the non-lone documents.

Now, $f_{d,j}^t \geq \frac{1}{C_m^2}f_{d,j}^*$, so $\frac{\tilde{f}_{d,i}}{f_{d,j}^t} \leq (1+\epsilon)C_m^2$. Also, $\gamma_{d,i}^t \leq 2\gamma_{d,i}^*$, since topic $i$ is the dominant in document $d$. Since $C_m \geq 2$, $\frac{\tilde{f}_{d,i}}{f_{d,j}^t}\gamma_{d,i}^t \leq (1+\epsilon)C_m^3\gamma_{d,i}^*$.

Also, note that $\sum_{d=1}^D \gamma_{d,i}^t \geq \frac{1}{C_m}\sum_{d=1}^D \gamma_{d,i}^*$, again, since $i$ is the dominant topic.

As usual, let's denote the set of lone documents $D_l$:

$$\beta_{i,j}^{t+1} \leq (1+\epsilon)C_m \frac{\sum_{d \in D_l} \beta_{i,j}^* \gamma_{d,i}^* + \sum_{d \in D \setminus D_l} C_m^3 \gamma_{d,i}^* \beta_{i,j}^t}{\sum_{d=1}^D \gamma_{d,i}^*}$$

As in the prior proofs, let's denote by $\alpha := \frac{\sum_{d \in D_l} \gamma_{d,i}^*}{\sum_{d=1}^D \gamma_{d,i}^*}$.

As in Lemma 4, $\alpha \geq 1 - o(1)$, so $\beta_{i,j}^{t+1} \leq (1+\epsilon)C_m(\alpha\beta_{i,j}^* + (1-\alpha)C_m^3\beta_{i,j}^t)$, which in turn implies that $\frac{\beta_{i,j}^{t+1}}{\beta_{i,j}^*} \leq (1+\epsilon)C_m(\alpha + (1-\alpha)C_m^3 C_\beta^t)$. In order to ensure that $\frac{\beta_{i,j}^{t+1}}{\beta_{i,j}^*} < \frac{C_\beta^t}{2}$, it would be sufficient to prove that

$$(1+\epsilon)C_m(\alpha + (1-\alpha)(C_m^3 C_\beta^t) < \frac{C_\beta^t}{2}$$

which is equivalent to $\alpha > \frac{C_m^3 C_\beta^t - \frac{C_\beta^t}{2(1+\epsilon)C_m}}{C_m^3 C_\beta^t - 1}$.

Let's look at the right hand side. As, by assumption, $C_\beta^t \geq 4C_m$, it follows that

$$\frac{C_m^3 C_\beta^t - \frac{C_\beta^t}{2(1+\epsilon)C_m}}{C_m^3 C_\beta^t - 1} \leq \frac{C_m^3 C_\beta^t - \frac{C_\beta^t}{2(1+\epsilon)C_m}}{C_m^3 C_\beta^t - \frac{C_\beta^t}{4C_m}}$$

Hence, the right hand side is upper bounded by

$$\frac{C_m^3 - \frac{1}{2(1+\epsilon)C_m}}{C_m^3 - \frac{1}{4C_m}} = 1 - \frac{\frac{\frac{2}{1+\epsilon}-1}{4C_m}}{C_m^3 - \frac{1}{4C_m}}$$

But, since $C_m$ is bounded by a constant, and $\alpha = 1 - o(1)$, the claim follows. $\qquad \square$

### 2.1.4  Upper bounds on the $\gamma$ values

Finally, we show that if we ever reach a point where the $\beta$ values are both upper and lower bounded by a constant, the $\gamma$ values one gets after the $\gamma$ step are appropriately upper bounded by a constant. More precisely:

**Lemma 6.** *Fix a particular document $d$. Let's assume the supports for the $\beta$ and $\gamma$ variables are correct. Furthermore, let $\frac{1}{C_m} \leq \frac{\beta_{i,j}^t}{\beta_{i,j}^*} \leq C_m$ for some constant $C_m$. Then, $\gamma_{d,i}^t \leq (1 + o(1))\gamma_{d,i}^*$.*

*Proof.* As in the proof of Lemma 3, let's look at the KKT conditions for $\gamma_{d,i}^t$ into a part corresponding to lone words $L_i$ and non-lone words. Multiplying 2.1 by $\gamma_{d,i}^t$ as before,

$$\gamma_{d,i}^t = \sum_{j \in L_i} \tilde{f}_{d,j} + \gamma_{d,i}^t \sum_{j \notin L_i} \frac{\tilde{f}_{d,j}}{f_{d,j}^t} \beta_{i,j}^t$$

Again, let $\tilde{\alpha} = \sum_{j \in L_i} \beta_{i,j}^*$.

By Lemma 3, certainly $\gamma_{d,i}^t \geq \frac{1}{C_m}\gamma_{d,i}^*$. Hence, $\frac{\tilde{f}_{d,j}}{f_{d,j}^t} \leq (1 + \epsilon)C_m^2$. So we have, $\gamma_{d,i}^t \leq (1 + \epsilon)(\tilde{\alpha}\gamma_{d,i}^* + C_m^3(1 - \tilde{\alpha})\gamma_{d,i}^t)$. In other words, this implies $\gamma_{d,i}^t \leq \frac{(1+\epsilon)\tilde{\alpha}}{1 - (1+\epsilon)C_m^3(1 - \tilde{\alpha})}\gamma_{d,i}^*$. Since $\tilde{\alpha} = 1 - o(1)$, it's easy to check that $\frac{\tilde{\alpha}}{1 - C_m^3(1 - \tilde{\alpha})} \leq 1 + o(1)$, which is enough for what we need. $\qquad \square$

So, as a corollary, we finally get:

**Corollary 7.** *For some $t_0 = O(\log(\frac{1}{\beta_{\min}^*})) = O(\log N)$ , it will be the case that for all $t \geq t_0$, $\frac{1}{C_\beta^0} \leq \frac{\beta_{i,j}^*}{\beta_{i,j}^t} \leq C_\beta^0$ for some constant $C_\beta^0$ and $\frac{1}{C_\gamma^0} \leq \frac{\gamma_{d,i}^*}{\gamma_{d,i}^t} \leq C_\gamma^0$ for some constant $C_\gamma^0$.*

This concludes Phase I of the analysis.

### 2.1.5  Phase II: Alternating minimization - upper and lower bound evolution

Taking Corollary 7 into consideration, we finally show that, if the $\beta$ and $\gamma$ values are correct up to a constant multiplicative factor, and we have the correct support, we can improve the multiplicative error in each iteration, thus achieving convergence to the correct values.

This portion bears resemblance to techniques like *state evolution* and *density evolution* in the literature for iterative methods for decoding error correcting codes. In those techniques, one keeps track of a certain quantity of the system that's evolving in each iteration. In density evolution, this is the probability density function of the messages that are being passed, in state evolution, it is a certain average and variance of the variables we are estimating.

In our case, we keep track of the "multiplicative accuracy" of our estimates $\gamma_{d,i}^t$, $\beta_{i,j}^t$. In particular, we will keep track of quantities $C_\gamma^t$ and $C_\beta^t$, such that at iteration $t$, $\frac{1}{C_\beta^t} \leq \frac{\beta_{i,j}*}{\beta_{i,j}^t} \leq C_\beta^t$ and $\frac{1}{C_\gamma^t} \leq \frac{\gamma_{d,i}*}{\gamma_{d,i}^t} \leq C_\gamma^t$ after the corresponding $\gamma$ iteration.

We will show that improvement in the quantities $C_\beta^t$ causes a large enough improvement in the $C_\gamma^t$ updates, so that after an alternating step of $\beta$ and $\gamma$ updates, $C_\beta^{t+1} \leq (C_\beta^t)^{1/2}$.

First, we show that when the $\beta$ variables are estimated up to a constant multiplicative factor, the constant for the $\gamma$ values after they've been iterated to convergence is slightly better than the constant for the $\beta$ values. More precisely:

**Lemma 8.** *Let's assume that our current iterates $\beta_{i,j}^t$ satisfy $\frac{1}{C_\beta^t} \leq \frac{\beta_{i,j}*}{\beta_{i,j}^t} \leq C_\beta^t$ for $C_\beta^t \geq \frac{1}{(1-\epsilon)^7}$. Then, after iterating the $\gamma$ updates to convergence, we will get values $\gamma_{d,i}^t$ that satisfy $(C_\beta^t)^{1/3} \leq \frac{\gamma_{d,i}*}{\gamma_{d,i}^t} \leq (C_\beta^t)^{1/3}$.*

*Proof.* As usual, we will split the KKT conditions for $\gamma_{d,i}^{t+1,t'}$ into two parts: one for the lone, and one for the non-lone words. Let's call the set of lone words $L_i$, as previously. Then. we have

$$\gamma_{d,i}^t = \sum_{j \in L_i} \tilde{f}_{d,j} + \gamma_{d,i}^t \sum_{j \notin L_i} \frac{\tilde{f}_{d,j}}{f_{d,j}^t} \beta_{i,j}^t$$

Again, let $\tilde{\alpha} = \sum_{j \in L_i} \beta_{i,j}^* = o(1)$, as we proved before.

Let's denote as $C_\gamma^t = \max_i(\max(\frac{\gamma_{d,i}^*}{\gamma_{d,i}^t}, \frac{\gamma_{d,i}^t}{\gamma_{d,i}^*}))$.

We claim that it has to hold that $C_\gamma^t \leq (C_\beta^t)^{1/3}$. Assume the contrary, and let $i_0 = \text{argmax}_i(\max(\frac{\gamma_{d,i}^*}{\gamma_{d,i}^t}, \frac{\gamma_{d,i}^t}{\gamma_{d,i}^*}))$.

Let's first assume that $\frac{\gamma_{d,i_0}^t}{\gamma_{d,i_0}^*} = C_\gamma^t$.

By the definition of $C_\gamma^t$,

$$\gamma_{d,i_0}^t = \sum_{j \in L_{i_0}} \tilde{f}_{d,j} + \gamma_{d,i_0}^t \sum_{j \notin L_{i_0}} \frac{\tilde{f}_{d,j}}{f_{d,j}^t} \beta_{i_0,j}^t \leq (1+\epsilon)(\tilde{\alpha}\gamma_{d,i_0}^* + (1-\tilde{\alpha})(C_\beta^t)^2(C_\gamma^t)^2 \gamma_{d,i_0}^*)$$

We claim that

$$(1+\epsilon)(\tilde{\alpha} + (1-\tilde{\alpha})(C_\beta^t)^2(C_\gamma^t)^2) \leq (C_\gamma^t)^{1/3} \tag{2.2}$$

which will be a contradiction to the definition of $C_\gamma^t$.

After a little rewriting, 2.2 translates to $\tilde{\alpha} \geq 1 - \frac{\frac{(C_\gamma^t)^{1/3}}{1+\epsilon}-1}{(C_\beta^t C_\gamma^t)^2 - 1}$. By our assumption on $C_\gamma^t$, $C_\beta^t \leq C_\gamma^3$, so the

right hand side above is upper bounded by $1 - \frac{\frac{(C_\gamma^t)^{1/3}}{1+\epsilon}-1}{(C_\gamma^t)^8 - 1}$.

But, Lemma 6 implies that certainly $C_\gamma^t \leq C_\gamma^0$, where $C_\gamma^0$ is some absolute constant. The function

$$f(c) = \frac{\frac{c^{1/3}}{1+\epsilon} - 1}{c^8 - 1}$$

can be easily seen to be monotonically decreasing on the interval of interest, and hence is lower bounded by $\frac{\frac{(C_\gamma^0)^{1/3}}{1+\epsilon}-1}{(C_\gamma^0)^8 - 1}$, which is in terms some absolute constant smaller than one. Since $\tilde{\alpha} = 1 - o(1)$. the claim we want is clearly true.

The case where $\frac{\gamma_{d,i_0}^*}{\gamma_{d,i_0}^t} = C_\gamma^t$ is similar. In this case,

$$\gamma_{d,i_0}^t = \sum_{j \in L_{i_0}} \tilde{f}_{d,j} + \gamma_{d,i_0}^t \sum_{j \notin L_{i_0}} \frac{\tilde{f}{d,j}}{f_{d,j}^t} \beta_{i_0,j}^t \geq (1-\epsilon)(\tilde{\alpha}\gamma_{d,i_0}^* + (1-\tilde{\alpha})\frac{1}{(C_\beta^t)^2(C_\gamma^t)^2}\gamma_{d,i_0}^*)$$

We then claim that

$$(1-\epsilon)(\tilde{\alpha} + (1-\tilde{\alpha})\frac{1}{(C_\beta^t)^2(C_\gamma^t)^2}) \geq \frac{1}{(C_\gamma^t)^{1/3}} \tag{2.3}$$

Again, 2.3 rewrites to:

$$\tilde{\alpha} \geq \frac{\frac{1}{(1-\epsilon)(C_\gamma^t)^{1/3}} - \frac{1}{(C_\beta^t)^2(C_\gamma^t)^2}}{1 - \frac{1}{(C_\beta^t)^2(C_\gamma^t)^2}} = 1 - \frac{1 - \frac{1}{(1-\epsilon)(C_\gamma^t)^{1/3}}}{1 - \frac{1}{(C_\beta^t C_\gamma^t)^2}}$$

Again, the right hand side above is upper bounded by $1 - \frac{1 - \frac{1}{(1-\epsilon)(C_\gamma^t)^{1/3}}}{1 - \frac{1}{(C_\gamma^t)^8}}$. But $C_\gamma \in [1, C_\gamma^0]$, and the

function $\frac{1 - \frac{1}{(1-\epsilon)c^{1/3}}}{1 - \frac{1}{c^8}}$ is monotonically increasing, so lower bounded by

$$\frac{1 - \frac{1}{(1-\epsilon)(\frac{1}{(1-\epsilon)^7})^{1/3}}}{1 - \frac{1}{(\frac{1}{(1-\epsilon)^7})^8}} = \frac{1 - (1-\epsilon)^{4/3}}{1 - (1-\epsilon)^{56}} \geq \frac{1}{42}$$

Hence, $1 - \dfrac{1 - \frac{1}{(1-\epsilon)(C_\gamma^t)^{1/3}}}{1 - \frac{1}{(C_\gamma^t)^{32}}}$ is upper bounded by $\frac{41}{42}$. Again, our bound on $\tilde{\alpha}$ gives us what we want.

$\square$

**Lemma 9.** *Let's assume that our current iterates $\beta_{i,j}^t$ satisfy $\frac{1}{C_\beta^t} \leq \frac{\beta_{i,j}*}{\beta_{i,j}^t} \leq C_\beta^t$, $C_\beta^t \geq \frac{1}{(1-\epsilon)^7}$, and after the corresponding $\gamma$ update, we get $\frac{1}{C_\gamma^t} \leq \frac{\gamma_{d,i}*}{\gamma_{d,i}^t} \leq C_\gamma^t$, where $C_\beta^t \geq (C_\gamma^t)^3$. Then, after one $\beta$ step, we will get new values $\beta_{i,j}^{t+1}$ that satisfy $\frac{1}{C_\beta^{t+1}} \leq \frac{\beta_{i,j}*}{\beta_{i,j}^{t+1}} \leq C_\beta^{t+1}$ where $C_\beta^{t+1} = (C_\beta^t)^{1/2}$.*

*Proof.* The proof proceeds in complete analogy with Lemmas 4 and 5.

Again, let's tackle the lower and upper bound separately. The upper bound condition is:

$$\alpha > \frac{(C_\beta^t C_\gamma^t)^2 - \frac{(C_\beta^t)^{1/2}}{(1+\epsilon)C_\gamma^t}}{(C_\gamma^t C_\beta^t)^2 - 1}$$

Using $C_\beta^t \geq (C_\gamma^t)^3$, we can upper bound the expression on the right by $1 - \dfrac{\frac{(C_\beta^t)^{1/6}}{1+\epsilon} - 1}{(C_\beta^t)^{8/3} - 1}$. The function

$f(c) = \dfrac{\frac{x^{1/6}}{1+\epsilon} - 1}{x^{8/3} - 1}$ is monotonically decreasing on the interval $[1, C_\beta^0]$ of interest, so because $\alpha = 1 - o(1)$, we get what we want.

Similarly, for the lower bound, we want that

$$\alpha > \frac{\frac{C_\gamma^t}{(C_\beta^t)^{1/2}(1-\epsilon)} - \frac{1}{(C_\gamma^t C_\beta^t)^2}}{1 - \frac{1}{(C_\beta^t C_\gamma^t)^2}}$$

Yet again, using $C_\beta^t \geq (C_\gamma^t)^3$, we get that the right hand side is upper bounded by

$$1 - \frac{1 - \frac{1}{(1-\epsilon)C_\beta^{1/6}}}{1 - \frac{1}{C_\beta^3}}$$

However, the function $f(c) = \dfrac{1 - \frac{1}{(1-\epsilon)c^{1/6}}}{1 - \frac{1}{c^{8/3}}}$ is monotonically increasing on the interval $[1, C_\beta^0]$, so lower bounded

by $\dfrac{1 - \frac{1}{(1-\epsilon)(\frac{1}{(1-\epsilon)^7})^{1/6}}}{1 - \frac{1}{(\frac{1}{(1-\epsilon)^7})^{8/3}}} = \dfrac{1 - (1-\epsilon)^{1/6}}{1 - (1-\epsilon)^{21}} \geq \frac{1}{126}$. Hence, $1 - \dfrac{1 - \frac{1}{(1-\epsilon)C_\beta^{1/6}}}{1 - \frac{1}{C_\beta^3}}$ is upper bounded by $\frac{125}{126}$, so using the fact

that $\alpha = 1 - o(1)$, we get what we want.

$\square$

Putting lemmas 8 and 9 together, we get:

**Lemma 10.** *Suppose it holds that $\frac{1}{C^t} \leq \frac{\beta_{i,j}*}{\beta_{i,j}^t} \leq C^t$, $C^t \geq \frac{1}{(1-\epsilon)^7}$. Then, after one KL minimization step with respect to the $\gamma$ variables and one $\beta$ iteration, we get new values $\beta_{i,j}^{t+1}$ that satisfy $\frac{1}{C^{t+1}} \leq \frac{\beta_{i,j}*}{\beta_{i,j}^{t+1}} \leq C^{t+1}$, where $C^{t+1} = \sqrt{C^t}$*

*Proof.* By Lemma 8, after the $\gamma$ iterations, we get $\gamma_{d,i}^t$ values that satisfy the condition $\frac{1}{(C')^t} \leq \frac{\gamma_{d,i}*}{\gamma_{d,i}^t} \leq (C')^t$, where $(C')^t = (C^t)^{1/3}$.

Then, by Lemma 9, after the $\gamma$ iteration, we will get $\frac{1}{C^{t+1}} \leq \frac{\beta_{i,j}*}{\beta_{i,j}^{t+1}} \leq C^{t+1}$, such that $C^{t+1} = (C^t)^{1/2}$, which is what we need.

$\square$

Hence, as a corollary, we get immediately:

**Corollary 11.** *Lemma 10 above implies that Phase III requires $O(\log(\frac{1}{\log(1+\epsilon')})) = O(\log(\frac{1}{\epsilon'}))$ iterations to estimate each of the topic-word matrix and document proportion entries to within a multiplicative factor of $1 + \epsilon'$.*

This finished the proof of Theorem 1 for the KL-tEM version of the updates. In the next section, we will remark on why the proofs are almost identical in the iterative and incomplete tEM version of the updates.

## 2.2 Iterative tEM updates, incomplete tEM updates

We show how to modify the proofs to show that the iterative tEM and incomplete tEM updates work as well. We'll just sketch the arguments as they are almost identical as above.

In those updates, when we are performing a $\gamma$ update, we initialize with $\gamma_{d,i}^t = 0$ whenever topic $i$ does not belong to document $d$, and $\gamma_{d,i}^t$ uniform among all the other topics.

Then, the way to modify Lemmas 3, 6, 8 is simple. Instead of arguing by contradiction about what happens at the KKT conditions, one will assume that at iteration $t'$ ($t'$ to indicate these are the separate iterations for the $\gamma$ variables that converge to the values $\gamma_{d,i}^t$) it holds that $\frac{1}{C_\gamma^{t'}}\gamma_{d,i}^* \leq \gamma_{d,i}^{t'} \leq C_\gamma^{t'}\gamma_{d,i}^*$. Then, as long as $C_\gamma^{t'}$ is too big, compared to $C_\beta^t$, one can show that $C_\gamma^{t'}$ is decreasing (to $C_\gamma^{t'+1} = (C_\gamma^t)^{1/2}$, say), using exactly the same argument we had before. Furthermore, the number of such iterations needed will clearly be logarithmic.

But the same argument as above proves the incomplete tEM updates work as well. Namely, even if we perform only one update of the $\gamma$ variables, they are guaranteed to improve.

## 2.3 Initialization

For completeness, we also give here a fairly easy, efficient initialization algorithm. Recall, the goal of this phase is to recover the supports - i.e. to find out which topics are present in a document, and identify the support of each topic. To reiterate the theorem statement:

**Theorem 12** (Restatement of Theorem 2). *If the number of documents is $\Omega(K^4 \log^2 K)$, there is a polynomial-time procedure which with probability $1 - \Omega(\frac{1}{K})$ correctly identifies the supports of the $\beta_{i,j}^*$ and $\gamma_{d,i}^*$ variables.*

We will find the topic supports first. Roughly speaking, we will devise a test, which will take as input two documents $d$, $d'$, and will try to determine if the two documents have a topic in common or not. The test will have no false positives, i.e. will never say NO, if the documents do have a topic in common, but might say NO even if they do. We will then, ensure that with high probability, for each topic we find a pair of documents intersecting in that topic, such that the test says YES.

We will also be able to identify which pairs intersect in exactly one topic, and from this we will be able to find all the topic supports. Having done all of this, finding the topics in each document will be easy as well. Roughly speaking, if a document doesn't contain a given topic, it will not contain all of the discriminative words in that document.

We give the algorithm formally as pseudocode Algorithm 1.

Now, let's proceed to analyze the above algorithm, proceeding in a few parts.

### 2.3.1 Constructing a no-false-positives test

First, we describe how one determines the supports of the topics. Let's define $Test(d, d') = \text{YES}$, if $\sum_j \min\{f_{d,j}^*, f_{d',j}^*\} \geq \frac{1}{2T}$, and NO otherwise. Then, we claim the following.

**Lemma 13.** *If $d, d'$ both contain a topic $i_0$, s.t. $\gamma_{d,i_0}^* \geq 1/T$, $\gamma_{d',i_0}^* \geq 1/T$ then $Test(d, d') = \text{YES}$. If $d, d'$ do not contain a topic $i_0$ in common, then $Test(d, d') = \text{NO}$.*

*Proof.* Let's prove the first claim.

$$\sum_j \min\{\tilde{f}_{d,j}, \tilde{f}_{d',j}\} \geq \sum_j (1-\epsilon)\min\{\beta_{i_0,j}^* \gamma_{d,i_0}^*, \beta_{i_0,j}^* \gamma_{d',i_0}^*\} \geq$$

---

**Algorithm 1** Initialization

---

**repeat** $K^4 \log^2 K$ times
    Sample a pair of documents $(d, d')$.
    ▷ *Test if $(d, d')$ intersect with no false positives:*
    **if** $\sum_j \min\{f^*_{d,j}, f^*_{d',j}\} \geq \frac{1}{2T}$ **then**
        $S_{d,d'} := \{j, s.t. f^*_{d,j}, f^*_{d',j} > 0\}$
        ▷ *"Weed-out" words that are not in the support of the intersection of (d,d')*
        **for** all documents $d'' \neq \{d, d'\}$ **do**
            **if** $\sum_j \min\{f^*_{d,j}, f^*_{d'',j}\} \geq \frac{1}{2T}$ and $\sum_j \min\{f^*_{d',j}, f^*_{d'',j}\} \geq \frac{1}{2T}$ **then**
                $S_{d,d'} = S_{d,d'} \cap j$, s.t $f^*_{d'',j} > 0$
            **end if**
        **end for**
    **end if**
**until**
▷ *Determine which $S_{a,b}$ correspond to documents intersecting in one topic only)*
**if** Set $S_{a,b}$ appears less than $D/K^{2.5}$ times, where $D$ is the total number of documents **then**
    Remove $S_{a,b}$.
**end if**
**if** Set $S_{a,b}$ can be written as the union of two other sets $S_{c,d}$, $S_{e,f}$, where neither is contained inside the other **then**
    Remove $S_{a,b}$.
**end if**
**if** Set $S_{a,b}$ is strictly contained inside $S_{d,d'}$ for some $S_{d,d'}$ **then**
    Remove $S_{d,d'}$.
**end if**
Remove duplicates.
The remaining lists $S_{a,b}$ are declared to be topic supports.

---

$$\sum_j (1-\epsilon)1/T\beta_{i_0,j}^* \geq 1/2T$$

Now, let's prove the second claim. Let's suppose $d, d'$ contain no topic in common.

Let's fix a topic $i_0$ that belongs to document $d$. By the "small discriminative words intersection", we have the following property:

$$\sum_{j \in i_0, j \in i'} \beta_{i,j}^* = o(1)$$

for any other topic $i' \neq i_0$.

Denoting by $T_{outside}$ the words belonging to topic $i_0$, and no topic in document $d'$, and $T_{inside}$ the words belonging to at least one other topic in $d'$, we have

$$\sum_{j \in T_{inside}} \beta_{i,j}^* \leq T \cdot o(1) = o(1)$$

For the words $j \in T_{outside}$, $\min\{f_{d,j}^*, f_{d',j}^*\} = 0$

By the above,

$$\sum_j \min\{\tilde{f}_{d,j}, \tilde{f}_{d',j}\} \leq (1+\epsilon)T^2 o(1) = o(1)$$

Thus, the test will say NO, as we wanted.

$\square$

### 2.3.2 Finding the topic supports from identifying pairs

Let's call $d, d'$ an *identifying pair* of documents for topic $i$, if $d, d'$ intersect in topic $i$ only, and furthermore the test says YES on that pair.

From this identifying pair, we show how to find the support of the topic $i$ in the intersection. What we'd like to do is just declare the words $j$, s.t. $f_{d,j}^*, f_{d',j}^*$ are both non-zero as the support of topic $i$. Unfortunately, this doesn't quite work. The reason is that one might find words $j$, s.t. they belong to one topic $i'$ in $d$, and another topic $i''$ in $d''$. Fortunately, this is easy to remedy. As per the pseudo-code above, let's call the following operation $WEEDOUT(d, d')$:

- Set $S = \{j, s.t. f_{d,j}^* > 0, f_{d',j}^* > 0\}$.

- For all $d''$, s.t. $Test(d, d'') = YES$, $Test(d', d'') = YES$:

- Set $S = S \cup \{j, s.t. f_{d'',j}^* > 0\}$

- Return $S$.

**Lemma 14.** *With probability $1 - \Omega(\frac{1}{K})$, for any pair of documents $d, d'$ intersecting in one topic, $WEEDOUT(d, d')$ is the support of $S$.*

*Proof.* For this, we prove two things. First, it's clear that $S$ is initialized in the first line in a way that ensures that it contains all words in the support of topic $i$. Furthermore, it's clear that at no point in time we will remove a word $j$ from $S$ that is in the support of topic $i$. Indeed - if $Test(d, d'') = YES$ and $Test(d', d'') = YES$, then by Lemma 13 document $d''$ must contain topic $i$. In this case, $f_{d'',j}^* > 0$, and we won't exclude $j$ from $S$.

So, we only need to show that the words that are not in the support of topic $i$ will get removed.

Let $d, d'$ intersect in a topic $i$. Let a word $j$ be outside the support of a given topic $i$. Because of the independent topic inclusion property, the probability that a document $d''$ contains topic $i$, and no other topic containing $j$ is $\Omega(1/K)$.

Since the number of documents is $\Omega(K^4 \log^2 K)$, by Chernoff, the probability that there is a document $d''$, s.t. $Test(d, d'') = YES$, $Test(d', d'') = YES$, but $f_{d'',j}^* = 0$, is $1 - \Omega(\frac{1}{e^{K^2 \log^2 K}})$. Union bounding over all words $j$, as well as pairs of documents $d, d'$, we get that for any documents $d, d'$ intersection in a topic $i$, we get the claim we want.

$\square$

### 2.3.3   Finding the identifying pairs

Finally, we show how to actually find the identifying pairs. The main issue we need to handle are documents that do intersect, and the TEST returns yes, but they intersect in more than one topic. There's two ingredients to ensuring this is true in the above algorithm.

- First, we delete all sets in the list of sets $S_{a,b}$ that show up less than $D^2/K^{2.5}$ number of times.

- Second, we remove sets that can be written as the union of two other sets $S_{c,d}$, $S_{e,f}$, where neither of the two is contained inside the other.

- After this, we delete the non-maximal sets in the list.

The following lemma holds:

**Lemma 15.** *Each topic has $\Omega(D^2/K^2)$ identifying pairs with probability $1 - \Omega(\frac{1}{K})$.*

*Proof.* Let $\mathcal{I}_i$ be the event that there are at least $\Omega(D^2/k^2)$ identifying pairs for topic $i$. Let $N_i$ be a random variable denoting the number of documents which have topic $i$ as a dominating topic. Furthermore, let $\mathcal{M}_i$ be the event that there are at least $\frac{N_i^2}{2} - K\sqrt{N_i^2}$ identifying pairs among the $N_i$ ones that have $i$ as a dominating topic. By the dominant topic equidistribution property, probability that a document $d$ has a topic $i$ as a dominating topic is at least $C/K$ for some constant $C$. Then, clearly,

$$\Pr[\cap_{i=1}^K \mathcal{I}_i] \geq \Pr\left[\cap_{i=1}^K \left(N_i \geq \frac{1}{2}C\frac{D}{K}\right)\right] \Pr\left[\cap_{i=1}^K \mathcal{M}_i | \cap_{i=1}^K \left(N_i \geq \frac{1}{2}C\frac{D}{K}\right)\right]$$

Let's estimate $\Pr\left[\cap_{i=1}^K \left(N_i \geq \frac{1}{2}C\frac{D}{K}\right)\right]$ first. The probabilities that different documents have $i_0$ as the dominating topic are clearly independent, so by Chernoff, if $N_i$ is the number of documents where $i$ is the dominating topic,

$$\Pr[N_i \geq (1-\epsilon)C\frac{D}{K}] \geq 1 - e^{-\frac{\epsilon^2}{3}C\frac{D}{K}}$$

Since $D = \Omega(K^2)$, plugging in $\epsilon = \frac{1}{2}$, $\Pr[N_i < \frac{1}{2}C\frac{D}{K}] \geq 1 - e^{-\Omega(K)}$. Union bounding over all topics, we get that with probability $\Pr\left[\cap_{i=1}^K \left(N_i \geq \frac{1}{2}C\frac{D}{K}\right)\right] \geq 1 - \frac{1}{K}$.

Now, let's consider $\Pr\left[\cap_{i=1}^K \mathcal{M}_i | \cap_{i=1}^K \left(N_i \geq \frac{1}{2}C\frac{D}{K}\right)\right]$. The event $\cap_{i=1}^K \left(N_i \geq \frac{1}{2}C\frac{D}{K}\right)$ can be written as the disjoint union of events

$$\{\mathbb{D} = \cup_{i=1}^K D_i, \forall i \neq j, D_i \cap D_j = \emptyset\}$$

where $\mathbb{D}$ is the set of all documents, $D_i$ is the set of documents that have $i$ as the dominating topic, and $|D_i| \geq \frac{1}{2}C\frac{D}{K}, \forall i$. (i.e. all the partitions of $\mathbb{D}$ into $K$ sets of sufficiently large size). Evidently, if we prove a lower bound on $\Pr\left[\cap_{i=1}^K \mathcal{M}_i | E\right]$ for any such event $E$, it will imply a lower bound on $\Pr\left[\cap_{i=1}^K \mathcal{M}_i | \cap_{i=1}^K \left(N_i \geq \frac{1}{2}C\frac{D}{K}\right)\right]$. For any such event, consider two documents $d, d' \in \{D_i\}$, i.e. having $i$ as the dominating topic. Let $\mathcal{I}_{d,d'}$ be an indicator variable denoting the event that $d, d'$ do not intersect in an additional topic. $\Pr[\mathcal{I}_{d,d'} = 1] = 1 - o(1)$, by the independent topic inclusion property and the events $\mathcal{I}_{d,d'}$ are easily seen to be pairwise independent. Furthermore, $\text{Var}[\mathcal{I}_{d,d'}] = o(1)$. By Chebyshev's inequality,

$$\Pr\left[\sum_{d,d' \in D_i} \mathcal{I}_{d,d'} \geq \frac{1}{2}D_i^2 - c\sqrt{D_i^2}\right] \geq 1 - \frac{1}{c^2}$$

If $N_i = \Omega(K\log K)$, plugging in $c = K$, we get that $\Pr\left[\sum_{d,d' \in D_i} \mathcal{I}_{d,d'} = \Omega(D_i^2)\right] \geq 1 - \Omega(\frac{1}{K^2})$. Hence,

$\Pr\left[\cap_{i=1}^K \mathcal{M}_i | E\right] \geq 1 - \frac{1}{K}$, by a union bound, which implies $\Pr\left[\cap_{i=1}^K \mathcal{M}_i | \cap_{i=1}^K \left(N_i \geq \frac{1}{2}C\frac{D}{K}\right)\right] \geq 1 - \frac{1}{K}$.

Putting all of the above together, if $D = \Omega(K^2\log K)$, with probability $1 - \Omega(\frac{1}{K})$, all topics have $\Omega(D^2/K^2)$ identifying pairs, which is what we want.

$\square$

The lemma implies that with probability $1 - \Omega(\frac{1}{K})$, we will not eliminate the sets $S_{a,b}$ corresponding to topic supports.

We introduce the following concept of a "configuration". A set of words $C$ will be called a "configuration" if it can be constructed as the intersection of the discriminative words in some set of topics, i.e.

**Definition.** A set of words $C$ is called a configuration if there exists a set $I = \{I_1, \ldots, I_{|I|}\}$ of topics, s.t.

$$C = \cap_{i=1}^{|I|} W_{I_i}$$

Let's call the minimal size of a set $I$ that can produce $C$ the generator size of $C$.

Now, we claim the following fact:

**Lemma 16.** *If a configuration $C$ has generator size $\geq 3$, then with probability $1 - \Omega(\frac{1}{K})$, it cannot appear as one of the sets $S_{a,b}$ after step 2 in the WEEDOUT procedure.*

*Proof.* Since $C$ has generator size at least 3, if two sets $d, d'$ intersect in less than two topics, then step 1 in WEEDOUT cannot produce $S_{a,b}$ which is equal to $C$. Hence, prior to step 2, $C$ can only appear as $S_{d,d'}$ for $d, d'$ that intersect in at least 3 topics.

Let $\mathcal{I}_{d,d'}$ be an indicator variable denoting the fact that the pair of documents $d, d'$ intersects in at least 3 topics. We have $\Pr[\mathcal{I}_{d,d'} = 1] \leq 1/K^3 + 1/K^4 + \ldots 1/K^T = O(1/K^3)$ by the independent topic inclusion property.

If $\mathcal{I}_3$ is a variable denoting the total number of documents that intersect in at least 3 topics, again by Chebyshev as in Lemma 15 we get:

$$\Pr[\mathcal{I}_3 \geq \Theta(D/K^3) - c\Theta(\sqrt{D}/K^{3/2})] \geq 1 - \frac{1}{c^2}$$

Again, by putting $c = \sqrt{K}$, since the number of documents is $K^4 \log^2 K$, with probability $1 - \frac{1}{K}$, all configurations with generator size $\geq 3$ cannot appear as one of the sets $S_{a,b}$, as we wanted. $\qquad\square$

This means that after the WEEDOUT step, with probability $1 - \Omega(\frac{1}{K})$, we will just have sets $S_{a,b}$ corresponding to configurations generated by two topics or less. The options for these are severely limited: they have to be either a topic support, the union of two topic supports, or the intersection of two topic supports. We can handle this case fairly easily, as proven in the following lemma:

**Lemma 17.** *After the end of step 3, with probability $1 - \Omega(\frac{1}{K})$, the only remaining $S_{a,b}$ are those corresponding to topic supports.*

*Proof.* First, when we check if some $S_{d,d'}$ is the union of two other sets and delete it if yes, I claim we will delete the sets equal to configurations that correspond to unions of two topic supports (and nothing else). This is not that difficult to see: certainly the sets that do correspond to configurations of this type will get deleted.

On the other hand, if it's the case that $S_{a,b}$ corresponds to a single topic support, we won't be able to write it as the union of two sets $S_{d,d'}$, $S_{d'',d'''}$, unless one is contained inside the other - this is ensured by the existence of discriminative words.

Hence, after the first two passes, we will only be left with sets that are either topic supports, or intersections of two topic supports. Then, removing the non-maximal is easily seen to remove the sets that are intersections, again due to the existence of discriminative words. $\qquad\square$

### 2.3.4   Finding the document supports

Now, given the supports of each topic, for each document, we want to determine the topics which are non-zero in it. The algorithm is given in 2:

**Lemma 18.** *If a topic $i_0$ is such that $\gamma^*_{d,i_0} > 0$, it will be declared as "IN". If a topic $i_0$ is such that $\gamma^*_{d,i_0} = 0$, it will be declared as out.*

**Algorithm 2** Finding document supports

---

Initialize $R = \emptyset$.
**for** each $i$ **do**
    Compute $\text{Score}(i) = \sum_{j \in Support(i) \setminus R} \tilde{f}_{d,j}$
**end for**
Find $i^*$ such that $\text{Score}(i^*)$ is maximum.
**while** $\text{Score}(i^*) > 0$ **do**
    Output $i^*$ to be in the support of $d$.
    $R = R \cup support(i^*)$
    Recompute Score for every other topic.
    Find $i^*$ with maximum score.
**end while**

---

*Proof.* Consider a topic $i$. At any iteration of the while cycle, consider $\sum_{j \in Support(i) \setminus R} \tilde{f}_{d,j}$. Clearly, $\tilde{f}_{d,j} \geq (1 - \epsilon)\gamma_{d,i}^* \beta_{i,j}^*$. Also $\sum_{j \in R} \beta_{i,j}^* = To(1)$. Hence,

$$\sum_{j \in Support(i) \setminus R} \tilde{f}_{d,j} \geq (1 - \epsilon)\gamma_{d,i}^*(1 - To(1)) \geq \frac{1}{2}\gamma_{d,i}^*$$

So, topic $i$ will be added eventually.

On the other hand, let's assume the document doesn't contain a given topic $i_0$. Let's call $B$ the set of words $j$ which are in the support of $i_0$, and belong to at least one of the topics in document $d$. Then, $\sum_{j \in i_0} \tilde{f}_{d,j} = \sum_{j \in B} \tilde{f}_{d,j}$. Let $i^*$ be the topic which is present in the document but not added yet and has maximum value of $\gamma_{d,i}^*$. Then

$$\sum_{j \in B} \tilde{f}_{d,j} \leq (1 + \epsilon) \sum_{i \in d} \sum_{j \in B} \gamma_{d,i}^* \beta_{i,j}^* \leq$$

$$(1 + \epsilon)\gamma_{d,i^*}^* \sum_{i \in d} \sum_{j \in B} \beta_{i,j}^* \leq$$

$$(1 + \epsilon)T\gamma_{d,i^*}^* o(1) \leq \gamma_{d,i^*}^*] \cdot o(1)$$

Hence, topic $i^*$ will always get preference over $i_0$. Once all the topics which are present in the document have been added, it is clear that no more topic will be added since score will be 0.

$\square$

This finally finishes the proof of Theorem 12.

# 3 Case study 2: Dominating topics, seeded initialization

As a reminder, *seeded* initialization does the following:

- For each topic $i$, the user supplies a document $d$, in which $\gamma_{d,i}^* \geq C_l$.

- We initialize with $\beta_{i,j}^0 = f_{d,j}^*$.

The theorem we want to show is:

**Theorem 19** (Restatement of Theorem 3). *Given an instance of topic modelling satisfying the Case Study 2 properties specified above, where the number of documents is $\Omega(\frac{K \log^2 N}{\epsilon'^2})$, if we initialize with seeded initialization, after $O(\log(1/\epsilon') + \log N)$ of KL-tEM updates, we recover the topic-word matrix and topic proportions to multiplicative accuracy $1 + \epsilon'$.*

The proof will be in a few phases again:

- *Phase I: Anchor identification*: First, we will show that as long as we can identify the dominating topic in each of the documents, the anchor words will make progress, in the sense that after $O(\log N)$ number of rounds, the values for the topic-word estimates will be almost zero for the topics for which the word is not an anchor, and lower bounded for the one for which it is.

- *Phase II: Discriminative word identification*: Next, we show that as long as we can identify the dominating topics in each of the documents, and the anchor words were properly identified in the previous phase, the values of the topic-word matrix for words which do not belong to a certain topic will keep dropping until they reach almost zero, while being lower bounded for the words that do.

- For Phase I and II above, we will need to show that the dominating topic can be identified at any step. Here we'll leverage the fact that the dominating topic is sufficiently large, as well as the fact that the anchor words have quite a large weight.

- *Phase III: Alternating minimization*: Finally, we show that after Phase I and II above, we are back to the scenario of the previous section: namely, there is a "boosting" type of improvement in each next round.

## 3.1 Estimates on the dominating topic

Before diving into the specifics of the phases above, we will show what the conditions we need are to be able to identify the dominating topic in each of the documents. For notational convenience, let $\Delta_m$ be the $m$-dimensional simplex: $x \in \Delta_m$ iff $\forall i \in [m], 0 \le x_i \le 1$ and $\sum_i x_i = 1$.

First, during a $\gamma$ update, we are minimizing $KL(\tilde{f}_d || f_d)$ with respect to the $\gamma_d$ variables, so we need some way or arguing that whenever the $\beta$ estimates are not too bad, minimizing this quantity also quantifies how far the $\gamma_d$ variables are from $\gamma_d^*$.

Formally, we'll show the following:

**Lemma 20.** *If, for all $i$, $KL(\beta_i^* || \beta_i^t) \le R_\beta$, and $\min_{\gamma_d \in \Delta_K} KL(\tilde{f}_d || f_d) \le R_f$, after running a KL divergence minimization step with respect to the $\gamma_d$ variables, we get that $||\gamma_d^* - \gamma_d||_1 \le \frac{1}{p}(\sqrt{\frac{1}{2}R_\beta} + \sqrt{\frac{1}{2}R_f}) + \epsilon$.*

We will start with the following simple helper claim:

**Lemma 21.** *If the word-topic matrix $\beta$ is such that in each topic the anchor words have total probability at least $p$, then $||\beta^* v||_1 \ge p||v||_1$.*

*Proof.*
$$||\beta^* v||_1 = \sum_j |\sum_i \beta_{i,j}^* v_i| \ge \sum_i \sum_{j \in W_i} |\beta_{i,j}^* v_i| \ge \sum_i p|v_i| \ge p||v||_1$$

$\square$

**Lemma 22.** *If, for all $i$, $KL(\beta_i^* || \beta_i^t) \le R_\beta$, and $\min_{\gamma_d \in \Delta_K} KL(\tilde{f}_d || f_d) \le R_f$, after running a KL divergence minimization step with respect to the $\gamma_d$ variables, we get that $||\gamma_d^* - \gamma_d||_1 \le \frac{1}{p}(\sqrt{\frac{1}{2}R_\beta} + \sqrt{\frac{1}{2}R_f}) + \epsilon$.*

*Proof.* First, observe that $\min_{\gamma_d \in \Delta_K} KL(\tilde{f}_d || f_d) \le R_f$, at the the optimal $\gamma_d$, we have that $||\tilde{f}_d - f_d||_1^2 \le \frac{1}{2}R_f$, i.e. $||\tilde{f}_d - f_d|| \le \sqrt{\frac{1}{2}R_f}$, by Pinsker's inequality.

We will show that if $||\gamma_d^* - \gamma_d||_1$ is large, so must be $||\tilde{f}_d - f_d||_1$, and hence $KL(\tilde{f}_d || f_d)$ - which will contradict the above upper bound.

Let's consider $\beta^*$ as $N$ by $K$ matrix, and $\gamma^*$ and $f^*$ as $K$-dimensional vectors. Let $\beta^* \gamma^*$ just denote matrix-vector multiplication - so $f^* = \beta^* \gamma^*$. For any other vector $\hat{\gamma}$, let's denote $\hat{f} = \beta^t \tilde{\gamma}$. Then:

$$||\tilde{f} - \hat{f}||_1 = ||\tilde{f} - \beta^t \tilde{\gamma}||_1 = ||\tilde{f} - (\beta^* + (\beta^t - \beta^*))\tilde{\gamma}||_1 \ge$$

$$||\tilde{f} - \beta^* \tilde{\gamma}||_1 - ||(\beta^t - \beta^*)\tilde{\gamma}||_1 \tag{3.1}$$

Hence, $||\tilde{f} - \beta^*\tilde{\gamma}||_1 \le ||(\beta^t - \beta^*)\tilde{\gamma}||_1 + ||\tilde{f} - \hat{f}||_1$. However,
However,

$$||(\beta^t - \beta^*)\gamma||_1 \le \max_i \sum_j |\beta_{i,j}^t - \beta_{i,j}^*| \le \max_i \sqrt{\frac{1}{2}KL(\beta_i^*||\beta_i^t)} \le \sqrt{\frac{1}{2}R_\beta} \qquad (3.2)$$

The first inequality is a property of induced matrix norms, the second is via Pinsker's inequality.

So, by 3.1 and 3.2, $||\tilde{f} - \beta^*\tilde{\gamma}||_1 \le \sqrt{\frac{1}{2}R_\beta} + \sqrt{\frac{1}{2}R_f}$. But now, finally, Lemma 21 implies that $||\gamma_d^* - \gamma_d||_1 \le \frac{1}{p}(\sqrt{\frac{1}{2}R_f} + \sqrt{\frac{1}{2}R_\beta}) + \epsilon$.

$\square$

**Lemma 23.** *Suppose that for the dominating topic $i$ in a document $d$, $\gamma_{d,i}^* \ge C_l$, and for all other topics $i'$, $\gamma_{d,i'}^* \le C_s$, s.t. $C_l - C_s > \frac{1}{p}(\sqrt{\frac{1}{2}R_f} + \sqrt{\frac{1}{2}R_\beta}) + \epsilon$. Then, the above test identifies the largest topic. Furthermore, $\frac{1}{2}\gamma_{d,i}^* \le \gamma_{d,i}^t \le \frac{3}{2}\gamma_{d,i}^*$*

*Proof.* By Lemma 22, and the relationship between $l_1$ and total variation distance between distributions, we have that $|\gamma_{d,i}^t - \gamma_{d,i}^*| \le \frac{1}{2}\left(\frac{1}{p}\left(\sqrt{\frac{1}{2}R_f} + \sqrt{\frac{1}{2}R_\beta}\right) + \epsilon\right)$.

For the dominating topic $i$, $\gamma_{d,i}^t \ge C_l - \frac{1}{2}\left(\frac{1}{p}\left(\sqrt{\frac{1}{2}R_f} + \sqrt{\frac{1}{2}R_\beta}\right) + \epsilon\right)$. On the other hand, for any other topic $i'$, $\gamma_{d,i'}^t \le C_s + \frac{1}{2}\left(\frac{1}{p}\left(\sqrt{\frac{1}{2}R_f} + \sqrt{\frac{1}{2}R_\beta}\right) + \epsilon\right)$. Since $C_l - C_s \ge \frac{1}{p}\left(\sqrt{\frac{1}{2}R_f} + \sqrt{\frac{1}{2}R_\beta}\right) + \epsilon$, $\gamma_{d,i}^t > \gamma_{d,i'}^t$, so the test works.

On the other hand, since $\gamma_{d,i}^t \ge \gamma_{d,i}^* - \left(\frac{1}{p}\left(\sqrt{\frac{1}{2}R_f} + \sqrt{\frac{1}{2}R_\beta}\right) + \epsilon\right) \ge \gamma_{d,i}^* - \frac{1}{2}\gamma_{d,i}^* = \frac{1}{2}\gamma_{d,i}^*$. Similarly, $\gamma_{d,i}^t \le \gamma_{d,i}^* + \frac{1}{p}\left(\sqrt{\frac{1}{2}R_f} + \sqrt{\frac{1}{2}R_\beta}\right) + \epsilon \le \gamma_{d,i}^* + \frac{1}{2}\gamma_{d,i}^* = \frac{3}{2}\gamma_{d,i}^*$.

$\square$

## 3.2 Phase I: Determining the anchor words

We proceed as outlined. In this section we show that in the first phase of the algorithm, the anchor words will be identified - by this we mean that we will be able to show that if a word $j$ is an anchor for topic $i$, $\beta_{i,j}^t$ will be within a factor of roughly 2 from $\beta_{i,j}^*$, and $\beta_{i',j}^t$ will be almost 0 for any other topic $i'$.

We will assume throughout this and the next section that we can identify what the dominating topic is, and that we have an estimate of the proportion of the dominating topic to within a factor of 2. (We won't restate this assumption in all the lemmas in favor of readability.)

We will return to this issue after we've proven the claims of Phases I and II modulo this claim.

The outline is the following. We show that at any point in time, by virtue of the initialization, $\beta_{i,j}^t$ is pretty well lower bounded (more precisely it's at least constant times $\beta_{i,j}^*$). This enables us to show that $\beta_{i',j}^t$ will halve at each iteration - so in some polynomial number of iterations will be basically 0.

### 3.2.1 Lower bounds on the $\beta_{i,j}^t$ values

We proceed as outlined above. We show here that the $\beta_{i,j}^t$ variables are lower bounded at any point in time. More precisely, we show the following lemma:

**Lemma 24.** *Let $j$ be an anchor word for topic $i$, and let $i' \ne i$. Suppose that $\beta_{i',j}^t \le \beta_{i,j}^t$. Then, $\beta_{i,j}^{t+1} \ge (1 - \epsilon)C_l\beta_{i,j}^*$ holds.*

*Proof.* We'll prove a lower bound on each of the terms $\frac{\tilde{f}_{d,j}}{f_{d,j}^t}\beta_{i,j}^t$. Since the update on the $\beta$ variables is a convex combination of terms of this type, this will imply a lower bound on $\beta_{i,j}^{t+1}$.

For this, we upper bound $f_{d,j}^t$. We have:

$$f_{d,j}^t = \beta_{i,j}^t \gamma_{d,i}^t + \sum_{i' \ne i} \beta_{i',j}^t \gamma_{d,i'}^t$$

This means that $f_{d,j}^t$ is a convex combination of terms, each of which is at most $\beta_{i,j}^t$. Hence, $f_{d,j}^t \le \beta_{i,j}^t$ holds. But then $\frac{\tilde{f}_{d,j}}{f_{d,j}^t}\beta_{i,j}^t \ge \tilde{f}_{d,j} \ge (1-\epsilon)\beta_{i,j}^*\gamma_{d,i}^* \ge (1-\epsilon)C_l\beta_{i,j}^*$. This implies $\beta_{i,j}^{t+1} \ge (1-\epsilon)C_l\beta_{i,j}^*$, as we wanted.

$\square$

### 3.2.2 Decreasing $\beta_{i',j}^t$ values

We'll bootstrap to the above result. Namely, we'll prove that whenever $\beta_{i,j}^t \ge 1/C_\beta \beta_{i,j}^*$ for some constant $C_\beta$, the $\beta_{i',j}^t$ values decrease multiplicatively at each round. Prior to doing that, the following lemma is useful. It will state that whenever the values of the variables $\beta_{i',j}^t$ are somewhat small, we can get some reasonable lower bound on the values $\gamma_{d,i}^t$ we get after a step of KL minimization with respect to the $\gamma$ variables.

**Lemma 25.** *Let $j$ be an anchor for topic $i$, and let $i' \ne i$. Let $\beta_{i',j}^t \le b\beta_{i,j}^t$. Then, for any document $d$, when performing KL divergence minimization with respect to the variables $\gamma_d$, for the optimum value $\gamma_{d,i}^t$, it holds that $\gamma_{d,i}^t \ge (1-\epsilon)\frac{p}{1-b}\gamma_{d,i}^* - \frac{b}{1-b}$.*

*Proof.* The KKT conditions 2.1 imply that if we denote $A_i$ the set of anchors in topic $i$, $\sum_{j \in A_i} \frac{\tilde{f}_{d,j}}{f_{d,j}^t}\beta_{i,j}^t \le 1$. By the assumption of the lemma,
$$f_{d,j}^t \le b_{i,j}^t\gamma_{d,i}^t + bb_{i,j}^t(1-\gamma_{d,i}^t)$$
Since $\tilde{f}_{d,j} \ge (1-\epsilon)\beta_{i,j}^*\gamma_{d,i}^*$, this implies $\frac{\tilde{f}_{d,j}}{f_{d,j}^t}\beta_{i,j}^t \ge (1-\epsilon)\beta_{i,j}^*\frac{\gamma_{d,i}^*}{\gamma_{d,i}^t(1-b)+b}$, i.e. $\sum_{j \in A_i}(1-\epsilon)\beta_{i,j}^*\frac{\gamma_{d,i}^*}{\gamma_{d,i}^t(1-b)+b} \le 1$. Rearranging the terms, we get

$$\gamma_{d,i}^t \ge (1-\epsilon)\sum_{j \in A_i}\beta_{i,j}^*\frac{\gamma_{d,i}^*}{1-b} - \frac{b}{1-b} \ge (1-\epsilon)p\gamma_{d,i}^* - \frac{b}{1-b}$$

as we needed.

$\square$

With this in place, we show that the value $\beta_{i',j}^t$ when $j$ is an anchor for topic $i \ne i'$, decreases by a factor of 2 after the update for the $\beta$ variables.

This requires one more new idea. Intuitively, if we view the update as setting $\beta_{i',j}^{t+1}$ to $\beta_{i,j}^t$ multiplied by a convex combination of terms $\frac{f_{d,j}^*}{f_{d,j}^t}$, a large number of them will be zero, just because $f_{d,j}^* = 0$ unless topic $i$ belongs to document $d$.

By the topic equidistribution property then, the probability that this happens is only $O(1/K)$, so if the weight in the convex combination on these terms is reasonable, we will multiply $\beta_{i,j}^t$ by something less than 1, which is what we need.

Lemma 25 says that if $\gamma_{d,i}^*$ is reasonably large, we will estimate it somewhat decently. If $\gamma_{d,i}^*$ is small, then $f_{d,j}^*$ would be small anyway.

So we proceed according to this idea.

**Lemma 26.** *Let $j$ be an anchor for topic $i$. Let $\beta_{i',j}^t \le b\beta_{i,j}^t$ for $i' \ne i$, and let $\beta_{i,j}^t \ge 1/C_\beta\beta_{i,j}^*$ for some constant $C_\beta$. Then, $\beta_{i',j}^{t+1} \le b/2\beta_{i,j}^*$*

*Proof.* We will split the $\beta$ update as

$$\beta_{i',j}^{t+1} = \beta_{i',j}^t\left(\frac{\sum_{d \in D_1}\frac{\tilde{f}_{d,j}}{f_{d,j}^t}\gamma_{d,i'}^t}{\sum_d \gamma_{d,i'}^t} + \frac{\sum_{d \in D_2}\frac{\tilde{f}_{d,j}}{f_{d,j}^t}\gamma_{d,i'}^t}{\sum_d \gamma_{d,i'}^t} + \frac{\sum_{d \in D_3}\frac{\tilde{f}_{d,j}}{f_{d,j}^t}\gamma_{d,i'}^t}{\sum_d \gamma_{d,i'}^t}\right)$$

for some appropriately chosen partition of the documents into three groups $D_1, D_2, D_3$.

Let $D_1$ be documents which do not contain topic $i$ at all, $D_2$ documents which do contain topic $i$, and $\gamma_{d,i}^* \ge \frac{2b}{p}$, and $D_3$ documents which do contain topic $i$ and $\gamma_{d,i}^* < \frac{2b}{p}$.

The first part will just vanish because word $j$ is an anchord word for topic $i$, and topic $i$ does not appear in it, so $f_{d,j}^* = 0$ for all documents $d \in D_1$.

The second summand we will upper bound as follows. First, we upper bound $\frac{\tilde{f}_{d,j}}{f_{d,j}^t}$. We have that $f_{d,j}^t \geq \beta_{i,j}^t \gamma_{d,i}^t \geq 1/C_\beta \beta_{i,j}^* \gamma_{d,i}^t$. However, we can use Lemma 25 to lower bound $\gamma_{d,i}^t$. We have that $\gamma_{d,i}^t \geq (1-\epsilon)(\frac{p}{1-b}\gamma_{d,i}^* - \frac{b}{1-b}) \geq (1-\epsilon)\frac{p}{2(1-b)}\gamma_{d,i}^*$. This alltogether implies $\frac{\tilde{f}_{d,j}}{f_{d,j}^t} \leq \frac{1}{1-\epsilon}\frac{2(1-b)C_\beta}{p}$. Hence,

$$\beta_{i',j}^t \frac{\sum_{d\in D_2} \frac{\tilde{f}_{d,j}}{f_{d,j}^t}\gamma_{d,i'}^t}{\sum_d \gamma_{d,i'}^t} \leq \frac{1}{1-\epsilon}\frac{2C_\beta}{p}(1-b)\beta_{i',j}^t \frac{\sum_{d\in D_2}\gamma_{d,i'}^t}{\sum_d \gamma_{d,i'}^t}$$

Furthermore, $\sum_d \gamma_{d,i'}^t \geq \frac{1}{2}|D|C_l$. On the other hand, I claim $\sum_{d\in D_2}\gamma_{d,i'}^t = O(K/|D|)$. Recall that $D$ is the set of documents where topic $i'$ is the dominating topic - so by definition they contain topic $i$. On the other hand, if a document is in $D_2$ then it contains topic $i$ as well. However, by the independent topic inclusion property, the probability that a document with dominating topic $i'$ contains topic $i$ as well is $O(1/K)$. Hence,

$$\beta_{i',j}^t \frac{\sum_{d\in D_2} \frac{\tilde{f}_{d,j}}{f_{d,j}^t}\gamma_{d,i'}^t}{\sum_d \gamma_{d,i'}^t} = O(\frac{1}{K})b\beta_{i,j}^t$$

For the third summand we provide a trivial bound for the terms $\frac{\tilde{f}_{d,j}}{f_{d,j}^t}\beta_{i',j}^t\gamma_{d,i'}^t$:

$$\frac{\tilde{f}_{d,j}}{f_{d,j}^t}\beta_{i',j}^t\gamma_{d,i'}^t \leq (1+\epsilon)\beta_{i,j}^*\gamma_{d,i}^* \leq (1+\epsilon)\beta_{i,j}^*\frac{2b}{p}$$

Since again, $\sum_d \gamma_{d,i'}^t \geq \frac{1}{2}|D|C_l$, and again, the number of document in $D_3$ is at most $O(1/K)$ for the same reasons as before, we have that

$$\beta_{i',j}^t \frac{\sum_{d\in D_3} \frac{f_{d,j}^*}{f_{d,j}^t}\gamma_{d,i'}^t}{\sum_d \gamma_{d,i'}^t} \leq O(1/K)b\beta_{i,j}^* = O(1/K)b\beta_{i,j}^t$$

since $\beta_{i,j}^t \geq \frac{1}{C_\beta}\beta_{i,j}^*$.

From the above three bounds, we get that $\beta_{i',j}^{t+1} \leq O(1/K)b\beta_{i,j}^t \leq \frac{b}{2}\beta_{i,j}^t$.

$\square$

Now, we just have to put together the previous two claims: namely we need to show that the conditions for the decay of the non-anchor topic values, and the lower bound on the anchor-topic values are actually preserved during the iterations. We will hence show the following:

**Lemma 27.** *Suppose we initialize with seeded initialization. Then, after t rounds, if $j$ is an anchor word for topic $i$, $\beta_{i,j}^t \geq (1-\epsilon)C_l\beta_{i,j}^*$, and $\beta_{i',j}^t \leq 2^{-t}C_s\beta_{i,j}^*$.*

*Proof.* We prove this by induction.

Let's cover the base case first. In the seed document corresponding to topic $i$, $\gamma_{d,i}^* \geq C_l$, so at initialization $\beta_{i,j}^0 \geq C_l\beta_{i,j}^*$. On the other hand, if topic $i$ appears in the seed document for topic $i'$, then after initialization $\beta_{i',j}^0 \leq C_s\beta_{i,j}^* < \beta_{i,j}^0$. Hence, at initialization, the claim is true.

On to the induction step. If the claim were true at time step $t$, since $\beta_{i',j}^t \leq 2^{-t}C_s\beta_{i,j}^*$, by Lemma 24, $\beta_{i,j}^{t+1} \geq C_l\beta_{i,j}^*$ - so the lower bound still holds at time $t+1$. On the other hand, since $\beta_{i,j}^t \geq C_l\beta_{i,j}^*$, by Lemma 26, at time $t+1$, $\beta_{i',j}^t \leq 2^{-(t+1)}C_s\beta_{i,j}^*$.

Hence, the claim we want follows.

$\square$

Finally, we show the easy lemma that after the values $\beta_{i',j}^t$ have decreased to (almost) 0, $\beta_{i,j}^t \geq \frac{1}{2}\beta_{i,j}^*$.

**Lemma 28.** *Let word $j$ be an anchor word for topic $i$. Suppose $\beta_{i',j}^t \leq 2^{-t}C_s\beta_{i,j}^*$ and*

$$t > 10 \ \max(\log(N), \log(\frac{1}{\gamma_{\min}^*}), \log(\frac{1}{\beta_{\min}^*}))$$

*Then $4\beta_{i,j}^* \geq \beta_{i,j}^{t+1} \geq \frac{1}{4}\beta_{i,j}^*$.*

*Proof.* Let us do the lower bound first. It's easy to see $\sum_{i'} \beta_{i',j}^t \gamma_{d,i'} \leq 2\beta_{i,j}^t\gamma_{d,i}^t$. Hence,

$$\frac{\tilde{f}_{d,j}}{f_{d,j}^t}\beta_{i,j}^t\gamma_{d,i}^t = \frac{\tilde{f}_{d,j}}{\sum_{i'}\beta_{i',j}^t\gamma_{d,i'}^t}\beta_{i,j}^t\gamma_{d,i}^t \geq$$

$$\frac{1}{2}\frac{\tilde{f}_{d,j}}{\beta_{i,j}^t\gamma_{d,i}^t}\beta_{i,j}^t\gamma_{d,i}^t \geq (1-\epsilon)\frac{1}{2}\beta_{i,j}^*\gamma_{d,i}^*$$

Hence, after the update,

$$\beta_{i,j}^{t+1} \geq (1-\epsilon)\frac{1}{2}\beta_{i,j}^*\frac{\sum_d \gamma_{d,i}^*}{\sum_d \gamma_{d,i}^t} \geq \frac{1}{4}\beta_{i,j}^*$$

since $\gamma_{d,i}^t \leq 2\gamma_{d,i}^*$.

The upper bound is similar. Since $\sum_{i'} \beta_{i',j}^t \gamma_{d,i'} \geq \beta_{i,j}^t\gamma_{d,i}^t$,

$$\frac{\tilde{f}_{d,j}}{f_{d,j}^t}\beta_{i,j}^t\gamma_{d,i}^t \leq \tilde{f}_{d,j} \leq (1+\epsilon)\beta_{i,j}^*\gamma_{d,i}^*$$

Hence,

$$\beta_{i,j}^{t+1} \leq (1+\epsilon)\beta_{i,j}^*\frac{\sum_d \gamma_{d,i}^*}{\sum_d \gamma_{d,i}^t} \leq 2\beta_{i,j}^*$$

since $\gamma_{d,i}^t \geq \frac{1}{2}\gamma_{d,i}^*$. This certainly implies the claim we want.

$\square$

Furthermore, the following simple application of Lemma 25 is immediate and useful:

**Lemma 29.** *Let $t > \ 10\max(\log N, \log \frac{1}{\gamma_{\min}^*}, \log \frac{1}{\beta_{\min}^*})$. Then, $\gamma_{d,i}^t \geq \frac{p}{2}\gamma_{d,i}^*$.*

## 3.3 Discriminative words

We established in the previous section that after logarithmic number of steps, the anchor words will be correctly identified, and estimated within a factor of 2. We show that this is enough to cause the support of the discriminative words to be correctly identified too, as well as estimate them to within a constant factor where they are non-zero.

Same as before, we will assume in this section that we can identify the dominating topic.

We will crucially rely on the fact that the discriminative words will not have a very large dynamic range comparatively to their total probability mass in a topic. The high level outline will be similar to the case for the anchor words. We will prove that if a discriminative word $j$ is in the support of topic $i$, then $\beta_{i,j}^t$ will always be reasonably lower bounded, and this will cause the values $\beta_{i',j}^t$ to keep decaying for the topics $i'$ that the word $j$ does not belong to.

The reason we will need the bound on the dynamic range, and the proportion of the dominating topic, and the size of the dominating topic, is to ensure that the $\beta$'s are always properly lower bounded.

### 3.3.1 Bounds on the $\beta_{i,j}^t$ values

First, we show that because the discriminative words have a small range, the values $\beta_{i,j}^t$ whenever $\beta_{i,j}^*$ is non-zero are always maintained to be within some multiplicative constant (which depends on the range of the $\beta_{i,j}^*$).

As a preliminary, notice that having identified the anchor words correctly the $\gamma$ values are appropriately lower bounded after running the $\gamma$ update. Namely, by Lemma 29, $\gamma_{d,i}^t \geq p/2\gamma_{d,i}^*$

With this in hand, we show that the $\beta_{i,j}^t$ values are well upper bounded whenever $\beta_{i,j}^*$ is non-zero.

**Lemma 30.** *At any point in time $t$, $\beta_{i,j}^t \leq (1+\epsilon)\frac{2B}{C_l}\beta_{i,j}^*$.*

*Proof.* Since $\frac{\tilde{f}_{d,j}}{f_{d,j}^t}\beta_{i,j}^t\gamma_{d,i}^t \leq \tilde{f}_{d,j}$ we have:

$$\beta_{i,j}^{t+1} \leq \frac{\sum_d \tilde{f}_{d,j}}{\sum_d \gamma_{d,i}^t} \leq 2 \cdot \frac{\sum_d \tilde{f}_{d,j}}{\sum_d \gamma_{d,i}^*}$$

On the other hand, we claim that $\tilde{f}_{d,j} \leq (1+\epsilon)B\beta_{i,j}^*$. Indeed, $\tilde{f}_{d,j} \leq (1+\epsilon)\sum_i \gamma_{d,i}^*\beta_{i,j}^*$, and for any other topic $i'$, $\beta_{i',j}^* \leq B\beta_{i,j}^*$. Hence,

$$2 \cdot \frac{\sum_d \tilde{f}_{d,j}}{\sum_d \gamma_{d,i}^*} \leq \frac{2(1+\epsilon)DB\beta_{i,j}^*}{\sum_d \gamma_{d,i}^*}$$

However, since $\gamma_{d,i}^* \geq C_l$, the previous expression is at most

$$\frac{2(1+\epsilon)DB\beta_{i,j}^*}{DC_l} = \frac{2(1+\epsilon)B}{C_l}\beta_{i,j}^*$$

So, we get the claim we wanted.

$\square$

The lower bound on the $\beta_{i,j}^t$ values is a bit more involved. To show a lower bound on the $\beta_{i,j}^t$ values is maintained, we will make use of both the fact that the discriminative words have a small range, and that we have some small, but reasonable proportion of documents where $\gamma_{d,i}^* \geq 1 - \delta$. More precisely, we show:

**Lemma 31.** *Let $\beta_{i,j}^t \leq \frac{2(1+\epsilon)B}{C_l}\beta_{i,j}^*$ for all topics $i$ that word $j$ belongs to, and let $\beta_{i,j}^t \geq \frac{C_l}{B}\beta_{i,j}^*$. Then, $\beta_{i,j}^{t+1} \geq \frac{C_l}{B}\beta_{i,j}^*$ as well.*

*Proof.* Let's call $D_\delta$ the documents where $\gamma_{d,i}^* \geq 1 - \delta$. We can certainly lower bound

$$\beta_{i,j}^{t+1} \geq \frac{\sum_{d \in D_\delta} \frac{\tilde{f}_{d,j}}{f_{d,j}^t}\gamma_{d,i}^t\beta_{i,j}^t}{\sum_{d \in D} \gamma_{d,i}^t}$$

First, let's focus on $\frac{\tilde{f}_{d,j}}{f_{d,j}^t}\beta_{i,j}^t$. Then,

$$\tilde{f}_{d,j} \geq (1-\epsilon)(1-\delta)\beta_{i,j}^* \tag{3.3}$$

Furthermore, since $\sum_{d \in D_\delta} \gamma_{d,i}^t \geq \frac{1}{2}\sum_{d \in D_\delta} \gamma_{d,i}^*$ and $\sum_d \gamma_{d,i}^t \leq 2\sum_d \gamma_{d,i}^*$, we have that

$$\frac{\sum_{d \in D_\delta} \gamma_{d,i}^t}{\sum_d \gamma_{d,i}^t} \geq \frac{1}{4}\frac{8}{B}(1-\delta) = \frac{2}{B}(1-\delta) \tag{3.4}$$

Finally, we claim that $\frac{\beta_{i,j}^t}{f_{d,j}^t} \geq \frac{1}{2}$. Massaging this inequality a bit, we get it's equivalent to:

$$\frac{\beta_{i,j}^t}{f_{d,j}^t} \geq \frac{1}{2} \Leftrightarrow$$

$$f_{i,j}^t \le 2\beta_{i,j}^t \Leftrightarrow$$

$$\gamma_{d,i}^t \beta_{i,j}^t + \sum_{i'} \gamma_{d,i'}^t \beta_{i',j}^t \le 2\beta_{i,j}^t$$

The left hand side can be upper bounded by

$$\gamma_{d,i}^t \beta_{i,j}^t + \sum_{i'} \gamma_{d,i'}^t \frac{2(1+\epsilon)B^3}{C_l^2} \beta_{i,j}^t \le$$

$$\gamma_{d,i}^t \beta_{i,j}^t + (1 - \gamma_{d,i}^t) \frac{2(1+\epsilon)B^3}{C_l^2} \beta_{i,j}^t$$

by the assumptions of the lemma.

So, it is sufficient to show that $\gamma_{d,i}^t \beta_{i,j}^t + (1 - \gamma_{d,i}^t) \frac{2(1+\epsilon)B^3}{C_l^2} \beta_{i,j}^t \le 2\beta_{i,j}^t$, however this is equivalent after some rearrangement to $\gamma_{d,i}^t \ge 1 - \frac{1}{\frac{2(1+\epsilon)B^3}{C_l^2} - 1}$.

It's certainly sufficient for this that $\gamma_{d,i}^t \ge 1 - \frac{1}{\frac{B^3}{C_l^2}} = 1 - \frac{C_l^2}{B^3}$, but since since $\gamma_{d,i}^* \ge 1 - \delta$, by the definition of $\delta$ and Lemmas 22, 33, 34, this certainly holds.

Together with 3.4 and 3.3, we get that

$$\beta_{i,j}^{t+1} \ge (1 - \epsilon) \frac{2}{B} (1 - \delta)^2 \frac{1}{2} \beta_{i,j}^* \ge (1 - \epsilon) \frac{(1-\delta)^2}{B} \beta_{i,j}^*$$

But, by our assumptions, $(1 - \epsilon)(1 - \delta)^2 \ge C_l$, so the claim follows.

$\square$

### 3.3.2 Decreasing $\beta_{i',j}^t$ values

Finally, we show that if the discriminative word $j$ does not belong in topic $i'$, the value for $\beta_{i',j}^t$ will keep dropping. More precisely, the following is true:

**Lemma 32.** *Let word $j$ and topic $i$ be such that $\beta_{i',j}^* = 0$ and let $\beta_{i',j}^t \le b$. Furthermore, let for all the topics $i$ that $j$ belongs to hold: $\beta_{i,j}^t \ge 1/C_\beta \beta_{i,j}^*$ for some constant $C_\beta$. Finally, let $\gamma_{d,i}^t \ge \frac{1}{C_\gamma} \gamma_{d,i}^*$ for some constant $C_\gamma$. Then, $\beta_{i',j}^{t+1} \le b/2$.*

*Proof.* We proceed similarly as the analogous claim for anchor words. We split the update as

$$\beta_{i',j}^{t+1} = \beta_{i',j}^t \left( \frac{\sum_{d \in D_1} \frac{\tilde{f}_{d,j}}{f_{d,j}^t} \gamma_{d,i'}^t}{\sum_d \gamma_{d,i'}^t} + \frac{\sum_{d \in D_2} \frac{\tilde{f}_{d,j}}{f_{d,j}^t} \gamma_{d,i'}^t}{\sum_d \gamma_{d,i'}^t} \right)$$

for some appropriate partitioning of the documents $D_1, D_2$.

Namely, let $D_1$ be documents which do not contain any topic to which word $j$ belongs, the $D_2$ documents which contain at least one topic word $j$ belongs to.

For all the documents in $D_1$, $f_{d,j}^* = 0$, and we will provide a good bound for the terms $\frac{\tilde{f}_{d,j}}{f_{d,j}^t}$ in $D_2$, this way, we'll ensure $\beta_{i,j}^t$ gets multiplied by a quantity which is $o(1)$ to get $\beta_{i,j}^{t+1}$, which is of course enough for what we want.

Bounding the terms in $D_2$ is even simpler than before. We have:

$$f_{d,j}^t = \sum_i \beta_{i,j}^t \gamma_{d,i}^t \ge \frac{1}{C_\beta C_\gamma} \sum_i \beta_{i,j}^* \gamma_{d,i}^* = \frac{1}{C_\beta C_\gamma} f_{d,j}^*$$

Hence, $\frac{f_{d,j}^*}{f_{d,j}^t} \le C_\beta C_\gamma$.

Then we have:

$$\frac{\sum_d \frac{\tilde{f}_{d,j}}{f_{d,j}^t}\gamma_{d,i}^t}{\sum_d \gamma_{d,i}^t} \leq (1+\epsilon)\frac{\sum_d \frac{f_{d,j}^*}{f_{d,j}^t}\gamma_{d,i}^t}{\sum_d \gamma_{d,i}^t} \leq$$

$$4(1+\epsilon)\frac{\sum_d \frac{f_{d,j}^*}{f_{d,j}^t}\gamma_{d,i}^*}{\sum_d \gamma_{d,i}^*} \leq 4(1+\epsilon)\frac{\sum_{d \in D_2} C_\beta C_\gamma \gamma_{d,i}^*}{\sum_d \gamma_{d,i}^*}$$

But now, by the "weak topic correlation" property, $\frac{\sum_{d \in D_2}\gamma_{d,i}^*}{\sum_d \gamma_{d,i}^*} = o(1)$. Indeed, $D$ consists of the documents where $i'$ is the dominating topic. In order for the document to belong to $D_2$, at least one of the topics word $j$ belongs to must belong in the document as well. Since the word $j$ only belongs to $o(K)$ of the topics, and each document contains only a constant number of topics, by the small topic correlation property, the claim we want follows.

But then, clearly, $4\frac{\sum_{d \in D_2} C_\beta C_\gamma \gamma_{d,i}^*}{\sum_d \gamma_{d,i}^*} = o(1)$ as well.

Hence, $\beta_{i',j}^{t+1} = o(1)\beta_{i',j}^t \leq \frac{1}{2}\beta_{i',j}^t$, which is what we need. $\qquad \square$

## 3.4 Determining dominant topic and parameter range

To complete the proofs of the claims for Phase I and II, we need to show that at any point in time we correctly identify the dominant topic. Furthermore, in order to maintain the lower bounds on the estimates for the discriminative words, we will need to make sure that $\gamma_{d,i}^t$ is large as well in the documents where $\gamma_{d,i}^* \geq 1 - \delta$.

Let's proceed to the problem of detecting the largest topic first. By Lemma 23 all we need to do is bound $R_f$ and $R_\beta$ at any point in time during this phase. To do this, let's show the following lemma:

**Lemma 33.** *Suppose for the anchor words $\beta_{i,j}^t \geq C_1\beta_{i,j}^*$, for the discriminative words $\beta_{i,j}^t \geq C_2\beta_{i,j}^*$. Let $p_i$ be the proportion of anchor words in topic $i$. Then, $KL(\beta_i^*||\beta_i^t) \leq p_i \log(\frac{1}{C_1}) + (1 - p_i)\log(\frac{1}{C_2})$.*

*Proof.* This is quite simple. Since log is an increasing function,

$$KL(\beta_i^*||\beta_i^t) = \sum_j \beta_{i,j}^* \log(\frac{\beta_{i,j}^*}{\beta_{i,j}^t}) \leq p_i \log(\frac{1}{C_1}) + (1 - p_i)\log(\frac{1}{C_2})$$

$\qquad \square$

**Lemma 34.** *Suppose for the anchor words $\beta_{i,j}^t \geq C_1\beta_{i,j}^*$, for the discriminative words $\beta_{i,j}^t \geq C_2\beta_{i,j}^*$. Let $p_i$ be the proportion of anchor words in topic $i$. Then, $\min_{\gamma \in \Delta_K} KL(\tilde{f}_d||f_d) \leq \log(1+\epsilon) + \left(p \log(\frac{1}{C_1}) + (1 - p)\log(\frac{1}{C_2})\right)$.*

*Proof.* Also simple. The value of $KL(\tilde{f}_d||f_d)$ one gets by plugging in $\gamma_d = \gamma^*$ is exactly what is stated in the lemma.

$\qquad \square$

We'll just use the above two lemmas combined from our estimates from before. We know, for all the anchor words, that $\beta_{i,j}^t \geq C_l\beta_{i,j}^*$, and that for the discriminative words, $\beta_{i,j}^t \geq \frac{C_l}{B}\beta_{i,j}^*$. Hence, by Lemma 33, at any point in time $KL(\beta_i^*||\beta_i^t) \leq p \log(\frac{1}{C_l}) + (1 - p)\log(\frac{B}{C_l})$. So, by Lemma 23, it's enough that

$$C_l - C_s \geq \frac{1}{p}\left(\sqrt{2\left(p \log(\frac{1}{C_l}) + (1 - p)\log(BC_l)\right)} + \sqrt{\log(1+\epsilon)}\right) + \epsilon$$

.

Since $\frac{1}{p}\sqrt{2\left(p \log(\frac{1}{C_l}) + (1 - p)\log(BC_l)\right)} \leq \frac{1}{p}\sqrt{2\left(\log(\frac{1}{C_l}) + (1 - p)\log B\right)}$, to get a sense of the parameters one can achieve, for detecting the dominant topic, (ignoring $\epsilon$ contributions), it's sufficient that $C_l - C_s \geq \frac{2}{p}\sqrt{\max(\log(\frac{1}{C_l}), (1 - p)\log B)}$

If one thinks of $C_l$ as $1-\eta$ and $p \geq 1 - \frac{\eta}{\log B}$, since $\log(\frac{1}{C_l}) \approx \eta$ roughly we want that $C_l - C_s \gg \frac{2}{p}\sqrt{\eta}$. (One takeaway message here is that the weight we require to have on the anchors depends only *logarithmically* on the range $B$.)

Let's finally figure out what the topic proportions must be in the "heavy" documents. In these, we want $\gamma_{d,i}^* \geq 1 - \frac{C_l^2}{2B^3} + \frac{1}{p}\left(\sqrt{2\left(p\log(\frac{1}{C_l}) + (1-p)\log(BC_l)\right)} - \sqrt{\log(1+\epsilon)}\right) + \epsilon$. A similar approximation to the above gives that we roughly want $\gamma_{d,i}^* \geq 1 - \frac{1-2\eta}{2B^3} + \frac{2}{p}\sqrt{\eta}$.

## 3.5 Getting the supports correct

At the end of the previous section, we argued that after $O(\log N)$ rounds, we will identify the anchor words correctly, and the supports of the discriminative words as well. Furthremore, we will also have estimated the values of the non-zero discriminative word probabilities, as well the anchor word probabilities up to a multiplicative constant. Then, I claim that from this point onward at each of the $\gamma$ steps, the $\gamma^t$ values we get will have the correct support. Namely, the following is true:

**Lemma 35.** *Suppose for the anchor words and discriminative words $j$, if $\beta_{i,j}^* = 0$, it's true that $\beta_{i,j}^t = o(\frac{1}{n})$. Furthermore, suppose that if $\beta_{i,j}^* \neq 0$, $\frac{1}{C_\beta}\beta_{i,j}^* \leq \beta_{i,j}^t \leq C_\beta \beta_{i,j}^*$ for some constant $C_\beta$.*

*Then, when performing KL minimization with respect to the $\gamma$ variables, whenever $\gamma_{d,i}^* = 0$ we have $\gamma_{d,i}^t = 0$.*

*Proof.* Let $\gamma_{d,i}^* = 0$. If $\gamma_{d,i}^t \neq 0$, then the KKT conditions imply:

$$\sum_{j=1}^{N} \frac{\tilde{f}_{d,j}}{f_{d,j}^t}\beta_{i,j}^t = 1 \tag{3.5}$$

The only terms that are non-zero in the above summation are due to words $j$ that belong to at least one topic $i'$ in the document. Let $I$ be the set of words that belong to topic $i$ as well.

By Lemma 29, we know that $\gamma_{d,i}^t \geq p/2\gamma_{d,i}^*$ Since also $\beta_{i,j}^t \geq \frac{1}{C_\beta}\beta_{i,j}^*$, $f_{d,i}^t \geq \frac{p}{2C_\beta}f_{d,j}^*$. Since $\beta_{i,j}^t = o(\frac{1}{n})$ for words $j$ not in the support of topic $I$, $\sum_{j \notin I} \frac{\tilde{f}_{d,j}}{f_{d,j}^t}\beta_{i,j}^t = o(1)$.

On the other hand, for words in $I$, $\frac{\tilde{f}_{d,j}}{f_{d,j}^t}\beta_{i,j}^t \leq (1+\epsilon)\frac{2C_\beta^2}{p}\beta_{i,j}^*$, so $\sum_{j \in I} \frac{\tilde{f}_{d,j}}{f_{d,j}^t}\beta_{i,j}^t = o(1)$, by the small support intersection property.

However, this contradicts 3.5, so we get what we want.

$\square$

This means that after this phase, we will always correctly identify the supports of the $\gamma$ variables as well.

## 3.6 Alternating minimization

Now, finishing the proof of Theorem 19 is trivial. Namely, because of Lemmas 35, 27, and the analogue of 27, we are basically back to the case where we have the correct supports for both the $\beta$ and $\gamma$ variables. The only thing left to deal with is the fact that the $\beta$ variables are not quite zero.

Let $j$ be an anchor word for topic $i$. Let $\epsilon'' = 1 - (1-\epsilon')^{1/7}$. Similarly as in Lemma 29, for

$$t > 10\max(\log N, \log(\frac{1}{\epsilon''\gamma_{\min}^*}), \log(\frac{1}{\epsilon''\beta_{\min}^*}))$$

it holds that $\frac{f_{d,j}^*}{f_{d,j}^t} \geq (1-\epsilon')^{1/7}\frac{\beta_{i,j}^*\gamma_{d,i}^*}{\beta_{d,i}^t\gamma_{d,i}^t}$. The same inequality is true if $j$ is a lone word for topic $i$ in document $d$.

After the above event, the same proof from Case Study 1 implies that after $O(\log(\frac{1}{\epsilon'}))$ iterations we'll get

$$\frac{1}{1+\epsilon'}\beta_{i,j}^* \leq \beta_{i,j}^t \leq (1+\epsilon')\beta_{i,j}^*$$

and
$$\frac{1}{1+\epsilon'}\gamma^*_{i,j} \leq \gamma^t_{i,j} \leq (1+\epsilon')\gamma^*_{i,j}$$

This finishes the proof of Theorem 19.

# 4 Justification of prior assumptions

In this section we provide a brief motivation for our choice of properties on the topic model instances we are looking at. Nothing in the other sections crucially depends on this section, so it can be freely skipped upon first reading.

Most of our properties on the topic priors are inspired from what happens with the Dirichlet prior - specifically, variants of all of the "weak correlations" between topics hold for Dirichlet. Essentially the only difference between our assumptions and Dirichlet is the lack of smoothness. (Dirichlet is sparse, but only in the sense that it leads to a few "large" topics, but the other topics may be non-negligible as well.)

To the best of our knowledge, the lemmas proven here were not derived elsewhere, so we include them for completeness.

For all of the claims below, we will be concerned with the following scenario:

$\vec{\gamma} = (\gamma_1, \gamma_2, \ldots, \gamma_K)$ will be a vector of variables, and $\vec{\alpha} = (\alpha_1, \alpha_2, \ldots, \alpha_k)$ a vector of parameters. We will let $\vec{\gamma}$ be distributed as $\vec{\gamma} := Dir(\alpha_1, \alpha_2, \ldots, \alpha_k)$, where $\alpha_i = C_i/K^c$, for some constants $C_i$ and $c > 1$.

## 4.1 Sparsity

To characterize the sparsity of the topic proportions in a document, we will need the following lemma from (Telgarsky, 2013):

**Lemma 36.** *(Telgarsky, 2013) For a Dirichlet distribution with parameters $(C_1/k^c, C_2/k^c, \ldots, C_k/k^c)$, the probability that there are more than $c_0 \ln k$ coordinates in the Dirichlet draw that are $\geq 1/k^{c_0}$ is at most $1/k^{c_0}$.*

It's clear how this is related to our assumption: if one considers the coordinates $\geq \frac{1}{k^{c_0}}$ as "large", we assume, in a similar way, that there are only a few "large" coordinates. The difference is that we want the rest of the coordinates to be exactly zero.

## 4.2 Weak topic correlations

We will prove that the Dirichlet distribution satisfies something akin to the *weak topic correlations* property. We prove that when conditioning on some small $(o(K))$ set of topics being small, the marginal distributions for the rest of the topic proportions are very close to the original ones. This implies our "weak topic correlations" property.

The following is true:

**Lemma 37.** *Let $\vec{\gamma} = (\gamma_1, \gamma_2, \ldots, \gamma_K)$ be distributed as specified above.*

*Let $S$ be a set of topics of size $o(K)$, and let's denote by $\gamma_S$ the vector of variables corresponding to the topics in the set $S$, and $\gamma_{\bar{S}}$ the rest of the coordinates. Furthermore, let's denote by $\tilde{\gamma}_{\bar{S}}$ the distribution of $\gamma_{\bar{S}}$ conditioned on all the coordinates of $\gamma_S$ being at most $1/K^{c_1}$ for $c_1 > 1$.*

*Then, for any $i \in \bar{S}$ and $\gamma = 1 - \delta$, any $\delta = \Omega(1)$,*
*$\mathbb{P}_{\gamma_{\bar{S}}}(\gamma_i = \gamma) = (1 \pm o(1))\mathbb{P}_{\tilde{\gamma}_{\bar{S}}}(\gamma_i = \gamma)$.*

*Proof.* It's a folklore fact that if $\vec{Y} = \text{Dir}(\vec{\alpha})$, then

$$(Y_1, Y_2, \ldots, Y_{i-1}, Y_{i+1}, \ldots, Y_K | Y_i = y_i) = (1 - y_i)Dir(\alpha_1, \alpha_2, \ldots, \alpha_{i-1}, \alpha_{i+1}, \ldots, \alpha_K)$$

Applying this inductively, we get that $\tilde{\gamma}_{\bar{S}} = (1 - \sum_{j \in S} \gamma_i)\text{Dir}(\vec{\alpha}_{\bar{S}})$. Let's denote $s := \sum_{j \in S} \gamma_i$, and $\tilde{s} = \sum_{i \in S} \alpha_i$. Then, since $\gamma_i \leq 1/K^{c_1}$ for $i \in S$, $s = o(1)$. Similarly, $\tilde{s} = o(1)$.

For notational convenience, let's call $\tilde{\alpha}_0 = \sum_{i \notin S} \alpha_i$, and $\alpha_0 = \sum_i \alpha_i = \tilde{\alpha}_0 + \tilde{s}$.

The marginal distribution of variable $Y_i$ where $\vec{Y} = \text{Dir}(\vec{\alpha})$ is $\text{Beta}(\alpha_i, \alpha_0 - \alpha_i)$.
Hence,

$$\mathbb{P}_{\gamma_{\bar{S}}}(\gamma_i = \gamma) = \frac{1}{B(\alpha_i, \tilde{\alpha}_0 + \tilde{s} - \alpha_i)} \gamma^{\alpha_i - 1}(1 - \gamma)^{\tilde{\alpha}_0 + \tilde{s} - \alpha_i - 1}$$

and

$$\mathbb{P}_{\tilde{\gamma}_{\bar{S}}}(\gamma_i = \gamma) = \frac{1}{B(\alpha_i, \tilde{\alpha}_0 - \alpha_i)} \left(\frac{\gamma}{1-s}\right)^{\alpha_i - 1}\left(1 - \frac{\gamma}{1-s}\right)^{\tilde{\alpha}_0 - \alpha_i - 1}$$

The following holds:

$$\frac{\gamma^{\alpha_i - 1}(1 - \gamma)^{\tilde{\alpha}_0 + \tilde{s} - \alpha_i - 1}}{(\frac{\gamma}{1-s})^{\alpha_i - 1}(1 - \frac{\gamma}{1-s})^{\tilde{\alpha}_0 - \alpha_i - 1}} =$$

$$(1-s)^{\alpha_i - 1}\left(\frac{(1-s)(1-\gamma)}{1-s-\gamma}\right)^{-\alpha_i - 1}(1-\gamma)^{\tilde{s}} =$$

$$\left(1 + \frac{s}{1 - s - \gamma}\right)^{-\alpha_i - 1}(1-\gamma)^{\tilde{s}}$$

Now, I claim the above expression is $1 \pm o(1)$.

We'll just prove this for each of the terms individually. Since $1 + \frac{s}{1-s-\gamma} \geq 1$ and $-1 - \alpha_i \leq -1$, it follows that $(1 + \frac{s}{1-s-\gamma})^{-\alpha_i - 1} \leq 1$. On the other hand, by Bernoulli's inequality, $(1 + \frac{s}{1-s-\gamma})^{-\alpha_i - 1} \geq 1 - (\alpha_i + 1)\frac{s}{1-s-\gamma} \geq 1 - o(1)$, since $\gamma = 1 - \delta$, for some constant $\delta$, by our assumptions.

For the second term, since $1 - \gamma \leq 1$ and $\tilde{s} \geq 0$, $(1-\gamma)^{\tilde{s}} \leq 1$. On the other hand, again by Bernoulli's inequality, $(1-\gamma)^{\tilde{s}} \geq 1 - \gamma\tilde{s} = 1 - o(1)$, as we needed.

Comparing $B(\alpha_i, \tilde{\alpha}_0 + \tilde{s} - \alpha_i)$ and $B(\alpha_i, \tilde{\alpha}_0 - \alpha_i)$ is not so much more difficult. By definition, $B(\alpha_i, \alpha_0 - \alpha_i) = \int_0^1 x^{\alpha_i - 1}(1 - x)^{\alpha_0 - \alpha_i - 1} dx$, so

$$\frac{B(\alpha_i, \tilde{\alpha}_0 + \tilde{s} - \alpha_i)}{B(\alpha_i, \alpha_0 - \alpha_i)} =$$

$$\frac{\int_0^1 x^{\alpha_i - 1}(1 - x)^{\tilde{\alpha}_0 + \tilde{s} - \alpha_i - 1} dx}{\int_0^1 x^{\alpha_i - 1}(1 - x)^{\tilde{\alpha}_0 - \alpha_i - 1} dx}$$

We'll just bound each of the ratios

$$\frac{x^{\alpha_i - 1}(1 - x)^{\tilde{\alpha}_0 + \tilde{s} - \alpha_i - 1}}{x^{\alpha_i - 1}(1 - x)^{\tilde{\alpha}_0 - \alpha_i - 1}}$$

Namely, this is just $(1 - x)^{\tilde{s}}$. Same as above, $1 - o(1) \leq (1 - \gamma)^{\tilde{s}} \leq 1$. Hence, these are within a constant from each other.

$\square$

## 4.3 Dominant topic equidistribution

Now, we pass to proving a smooth version of the dominant topic equidistribution property. Namely, for a threshold $x_0 = o(1)$, we can consider a topic "large" whenever it's bigger than $x_0$. We will show that for any topics $Y_i$, $Y_j$, the probabilities that $Y_i > x_0$ and $Y_j > x_0$ are within a constant from each other.

Mathematically formalizing the above statement, we will prove the following lemma:

**Lemma 38.** *Let $\vec{\gamma} = (\gamma_1, \gamma_2, \ldots, \gamma_K)$ be distributed as specified above. Then, $\frac{\mathbb{P}(Y_i > x_0)}{\mathbb{P}(Y_j > x_0)} = O(1)$, for any $i, j$ if $x_0 = o(1)$.*

*Proof.* As before, the marginal distribution of $Y_i$ is Beta$(\alpha_i, \alpha_0 - \alpha_i)$. The Beta distribution pdf is just $\mathbb{P}(x) = \frac{x^{\alpha_i-1}(1-x)^{\alpha_0-\alpha_i-1}}{B(\alpha_i, \alpha_0-\alpha_i)}$, where $B(\alpha_i, \alpha_0 - \alpha_i) = \int_0^1 x^{\alpha_i-1}(1-x)^{\alpha_0-\alpha_i-1}\, dx$.

Hence, the ratio we care about can be written as

$$\frac{(\int_{x_0}^1 x^{\alpha_i-1}(1-x)^{\alpha_0-\alpha_i-1}\, dx)/B(\alpha_i, \alpha_0-\alpha_i)}{(\int_{x_0}^1 x^{\alpha_j-1}(1-x)^{\alpha_0-\alpha_j-1}\, dx)/B(\alpha_j, \alpha_0-\alpha_j)}$$

To get a bound on this ratio, it's sufficient to bound the normalization constants $B(\alpha_i, \alpha_0 - \alpha_i)$ and $B(\alpha_j, \alpha_0 - \alpha_j)$, as well as the ratio $\frac{\int_{x_0}^1 x^{\alpha_i-1}(1-x)^{\alpha_0-\alpha_i-1}\, dx}{\int_{x_0}^1 x^{\alpha_j-1}(1-x)^{\alpha_0-\alpha_j-1}\, dx}$. Let's prove first that $B(\alpha_i, \alpha_0-\alpha_i) \simeq \mathbb{B}(\alpha_j, \alpha_0 - \alpha_j)$

By definition, $B(\alpha_i, \alpha_0 - \alpha_i) = \int_0^1 x^{\alpha_i-1}(1-x)^{\alpha_0-\alpha_i-1}\, dx$. The way we'll analyze this quantity is that we'll divide the integral in two parts, one from 0 to $\frac{1}{2}$ and one from $\frac{1}{2}$ to 1.

Since $\alpha_0 = O(1)$, it follows that $\alpha_0 - \alpha_i - 1 \gtrsim -1$ and $\alpha_0 - \alpha_i - 1 \lesssim 1$. Hence, $(1-x)^{\alpha_0-\alpha_i-1} = \Theta(1)$. It follows that

$$\int_0^{\frac{1}{2}} x^{\alpha_i-1}(1-x)^{\alpha_0-\alpha_i-1}\, dx \simeq \int_0^{\frac{1}{2}} x^{\alpha_i-1}\, dx =$$

$$\simeq \frac{(1/2)^{\alpha_i}}{\alpha_i} \simeq \frac{1}{\alpha_i}$$

where the last equality follows since $\frac{1}{2} \leq (1/2)^{\alpha_i} \leq 1$.

The second portion is not much more difficult. Since $\frac{1}{2} \leq \frac{1}{2}^{\alpha_i-1} \leq 1$, it follows

$$\int_{\frac{1}{2}}^1 x^{\alpha_i-1}(1-x)^{\alpha_0-\alpha_i-1}\, dx \simeq \int_{\frac{1}{2}}^1 (1-x)^{\alpha_0-\alpha_i-1}\, dx =$$

$$\simeq \frac{(1/2)^{\alpha_0-\alpha_i}}{\alpha_0 - \alpha_i} \simeq \frac{1}{\alpha_0}$$

where the last two equalities come about since $-1 \lesssim \alpha_0 - \alpha_i \lesssim 1$.

But the above two estimates proved that for any $i$, $B(\alpha_i, \alpha_0 - \alpha_i) \simeq \frac{1}{\alpha_i}$, as we needed.

So, we proceed onto bounding

$$\frac{\int_{x_0}^1 x^{\alpha_i-1}(1-x)^{\alpha_0-\alpha_i-1}\, dx}{\int_{x_0}^1 x^{\alpha_j-1}(1-x)^{\alpha_0-\alpha_j-1}\, dx}$$

We'll proceed in a similar fashion as before. We'll pick some point $x_T$, and if $x < x_T$, we will show that $x^{\alpha_j-1}(1-x)^{\alpha_0-\alpha_j-1}$ is within a constant factor from $x^{\alpha_i-1}(1-x)^{\alpha_0-\alpha_i-1}$. On the other hand, we will show that part of the integral where $x > x_T$ is dominated by the part where $x < x_T$, which will imply the claim we need.

Let's rewrite the ratio above a little:

$$\frac{x^{\alpha_j-1}(1-x)^{\alpha_0-\alpha_j-1}}{x^{\alpha_i-1}(1-x)^{\alpha_0-\alpha_i-1}} =$$

$$(\frac{x}{1-x})^{\alpha_j-\alpha_i} = e^{(\alpha_j-\alpha_i)\ln(\frac{x}{1-x})}$$

Proceeding as outlined, I claim that for sufficiently large constants $C_1, C_2$, s.t. if $x \leq 1 - \frac{1}{1+C_1 e^{\frac{1}{\alpha_i}\frac{1}{C_2}}}$, then $\frac{x^{\alpha_j-1}(1-x)^{\alpha_0-\alpha_j-1}}{x^{\alpha_i-1}(1-x)^{\alpha_0-\alpha_i-1}} = O(1)$. Let's call $x_T = 1 - \frac{1}{1+C_1 e^{\frac{1}{\alpha_i}\frac{1}{C_2}}}$.

The claim is then, that if $x_T \geq x \geq x_0$, that $(\alpha_j - \alpha_i)\ln(\frac{x}{1-x}) = O(1)$.

First let's assume, $\alpha_j - \alpha_i \geq 0$.

Then, if $\ln(\frac{x}{1-x}) < 0 \Leftrightarrow x < \frac{1}{2}$, the condition is of course satisfied. So let's assume $x \geq \frac{1}{2}$. When $\frac{1}{2} \leq x \leq x_T$, we get that $\frac{x}{1-x} \leq C_1 e^{e^{\frac{1}{\alpha_j}\frac{1}{C_2}}}$. Hence, $\ln(\frac{x}{1-x}) \leq \ln C_1 + \frac{1}{\alpha_j}\frac{1}{C_2}$. It follows that if $C_1, C_2$ are sufficiently large,

$$(\frac{x}{1-x})^{\alpha_j - \alpha_i} \leq e^{\ln(\frac{x}{1-x})\alpha_j} = O(1)$$

On the other hand, if $\alpha_j - \alpha_i \leq 0$, when $x \geq \frac{1}{2}$, $(\alpha_j - \alpha_i)\ln(\frac{x}{1-x}) \leq 0$, so we are fine. However, since $|\alpha_j - \alpha_i| \leq \alpha_i$, it's easy to check when $x \geq \frac{e^{-c_1/\alpha_i}}{1+e^{-c_1/\alpha_i}} > x_0$, that $(\alpha_j - \alpha_i)\ln(\frac{x}{1-x}) = O(1)$.

Finally, we want to claim that the portion of the integral from $x_T$ to 1 is dominated by the portion from $x_0$ to $x_T$.

We can show that the latter portion is $O(e^{-K})$, and the first is $\Omega(1)$.

Let's lower bound the first portion. We lower bound $\int_{x_0}^{x_T} x^{\alpha_i - 1}(1-x)^{\alpha_0 - \alpha_i - 1}\, dx$ by $x_T^{\alpha_i - 1}\int_{x_0}^{x_T}(1-x)^{\alpha_0 - \alpha_i - 1}\, dx$. For the first factor in the above expression, we use Bernoulli's inequality to prove it's $\Omega(1)$. For the second, the integral will evaluate to

$$\frac{(1-x_0)^{\alpha_0 - \alpha_i} - (1-x_T)^{\alpha_0 - \alpha_i}}{\alpha_0 - \alpha_i}$$

Let's lower bound the first term in the numerator. If $\alpha_0 - \alpha_i \geq 1$, another application of Bernoulli's inequality gives: $(1-x_0)^{\alpha_0 - \alpha_i} \geq 1 - (\alpha_0 - \alpha_i)x_0 \geq 1 - o(1)$. If, on the other hand, $0 \leq \alpha_0 - \alpha_i \leq 1$, $(1-x_0)^{\alpha_0 - \alpha_i} \geq 1 - x_0 \geq 1 - o(1)$.

Then, I claim that $(1-x_T)^{\alpha_0 - \alpha_i} = e^{-\Omega(K)}$. Indeed, for some constant $C_3$,

$$\left(\frac{1}{1+C_1 e^{\frac{1}{\alpha_j}\frac{1}{C_2}}}\right)^{\alpha_0 - \alpha_i} \leq \left(\frac{1}{C_3 e^{\frac{1}{\alpha_j}\frac{1}{C_2}}}\right)^{\alpha_0 - \alpha_i} =$$

$$= e^{-\ln(C_3 e^{\frac{1}{\alpha_j}\frac{1}{C_2}})(\alpha_0 - \alpha_i)}$$

However, since $\alpha_0 = \Omega(K\alpha_j)$ and $\alpha_0 - \alpha_i = \Omega(\alpha_0)$, the above expression is upper bounded by $e^{-\Omega(K)}$, which is what we were claiming. Hence, $x_T^{\alpha_i - 1}\int_{x_0}^{x_T}(1-x)^{\alpha_0 - \alpha_i - 1}\, dx = \Omega(1)$.

Let's upper bound the latter portion. This expression is upper bounded by

$$x_T^{\alpha_i - 1}\int_{x_T}^{1}(1-x)^{\alpha_0 - \alpha_i - 1}\, dx = x_T^{\alpha_i - 1}\frac{\left(\frac{1}{1+C_1 e^{\frac{1}{\alpha_j}\frac{1}{C_2}}}\right)^{\alpha_0 - \alpha_i}}{\alpha_0 - \alpha_i}$$

Now, we will separately bound each of $x_T^{\alpha_i - 1}$ and $\frac{\left(\frac{1}{1+C_1 e^{\frac{1}{\alpha_j}\frac{1}{C_2}}}\right)^{\alpha_0 - \alpha_i}}{\alpha_0 - \alpha_i}$.

The first term can be written as $\frac{1}{x_T^{1-\alpha_i}}$. Now, since $1 - \alpha_i \geq 0$, we can use Bernoulli's inequality to lower bound $x_T^{1-\alpha_i}$ by $1 - \frac{1}{1+C_1 e^{\frac{1}{\alpha_j}\frac{1}{C_2}}}(1-\alpha_i)$. Since $\frac{1}{1+C_1 e^{\frac{1}{\alpha_j}\frac{1}{C_2}}} = O(1/e^{\frac{1}{\alpha_j}})$, and $1 - \alpha_i \leq 1/2$, let's say, $1 - \frac{1}{1+C_1 e^{\frac{1}{\alpha_j}\frac{1}{C_2}}}(1-\alpha_i) = \Omega(1)$, i.e. $x_T^{\alpha_i - 1} = O(1)$.

For the second term, we already proved above that $(1-x_T)^{\alpha_0 - \alpha_i} = e^{-\Omega(K)}$, This implies that $\int_{x_T}^{1} x^{\alpha_i - 1}(1-x)^{\alpha_0 - \alpha_i - 1}\, dx = O(e^{-K})$, which finishes the proof.

$\square$

## 4.4 Independent topic inclusion

Finally, there's a very simple proxy for "independent topic inclusion". Again, as above, $\tilde{\gamma}_{\bar{S}} = (1 - \sum_{j \in S} \gamma_i)\mathrm{Dir}(\vec{\alpha}_{\bar{S}})$.

But, if we consider "inclusion" the probability that a given topic is "noticeable" (i.e. $\geq \frac{1}{n^{c_0}}$, say), we can use the above Lemma 38 to show that the probability that any topic is "large" (but still $o(1)$) is within a constant for all the topics in $\bar{S}$.

# 5 On common words

In this section, we show how one would modify the proofs from the previous section to handle common words as well. We stress that common words are easy to handle if one were allowed to filter them out, but we want to analyze under which conditions the variational inference updates could handle them on their own.

The difference in contrast to the previous sections is it's not clear how to argue progress for the common words: common words do not have lone documents. However, if we can't argue progress for the common words, then we can't argue progress for the $\gamma$ variables, so the entire argument seems to fail.

Formally, we consider the following scenario:

- On top of the assumptions we have either in Case Study 1 or Case Study 2, we assume that there are words which show up in all topics, but their probabilities are within a constant $\kappa$ from each other, $B \geq \kappa \geq 2$. We will call these *common* words. (The $\kappa \geq 2$ is without loss of generality. If the claim holds for a smaller $\kappa$, then it certainly holds for $\kappa = 2$. The only difference is that the estimates to follow could be strengthened, but we assume $\kappa \geq 2$ to get cleaner bounds.)

- For each topic $i$, if $C$ is the set of common words, $\sum_{j \in C} \beta_{i,j}^* \leq \frac{1}{\kappa^{100}}$, i.e. there isn't too much mass on these words.

- Conditioned on topic $i$ being dominant, there is a probability of $1 - \frac{1}{\kappa^{100}}$ that the proportion of topic $i$ is at least $1 - \frac{1}{\kappa^{100}}$.

Then, recall the theorem we want to prove is:

**Theorem 39** (Restatement of Theorem 5)**.** *If we additionally have common words satisfying the properties specified above, after $O(\log(1/\epsilon') + \log N)$ of KL-tEM updates in Case Study 2, or any of the tEM variants in Case Study 1, and we use the same initializations as before, we recover the topic-word matrix and topic proportions to multiplicative accuracy $1 + \epsilon'$, if $1 + \epsilon' \geq \frac{1}{(1-\epsilon)^7}$.*

Our analysis here is fairly loose, since the result is anyway a little weak. (e.g. $1 - \frac{1}{\kappa^{100}}$ is not really the best value for the proportion of the dominating topic, or the proportion of such documents required.) At any rate, it will be clear from the proofs that the dependency of the dominating topic on $\kappa$ has to be of the form $1 - \frac{1}{\kappa^c}$, so it's not clear one would gain too much from the tightest possible analysis. The reason we are including this section is to show cases where our proof methods start breaking down.

We will do the proof for Case Study 1 first, after which Case Study 2 will easily follow.

## 5.1 Phase I with common words

The outline is the same as before. We prove the lower bounds on the $\gamma$ and $\beta$ variables first. Namely, we prove:

**Lemma 40.** *Suppose that the supports of $\beta$ and $\gamma$ are correct. Then, $\gamma_{d,i}^t \geq \frac{1}{2}\gamma_{d,i}^*$.*

*Proof.* Similarly as before, multiplying both sides of 2.1 by $\gamma_{d,i}^t$, we get that

$$\gamma_{d,i}^t \geq \sum_{L_i} \frac{f_{d,j}^*}{f_{d,j}^t} \beta_{i,j}^t \gamma_{d,i}^t \geq (1 - o(1))(1 - \frac{1}{\kappa^{100}})\gamma_{d,i}^* \geq \frac{1}{2}\gamma_{d,i}^*$$

where the second inequality follows since $1 - \frac{1}{\kappa^{100}}$ fraction of the words in topic $i$ is discriminative. $\square$

**Lemma 41.** *Suppose that the supports of the $\gamma$ and $\beta$ variables are correct. Additionally, if $i$ is a large topic in $d$, let $\frac{1}{2}\gamma_{d,i}^* \leq \gamma_{d,i}^t \leq 3\gamma_{d,i}^*$. Then, for a discriminative word $j$ for topic $i$, $\beta_{i,j}^{t+1} \geq \frac{1}{3}\beta_{i,j}^*$.*

*Proof.* Again, similarly as in Lemma 4,

$$\beta_{i,j}^{t+1} \geq \frac{\sum_{d \in D_l} (1 - \epsilon) \frac{\gamma_{d,i}^* \beta_{i,j}^*}{\gamma_{d,i}^t \cdot \beta_{i,j}^t} \beta_{i,j}^t \gamma_{d,i}^t}{\sum_{d=1}^{D} \gamma_{d,i}^t} =$$

$$(1-\epsilon)\beta_{i,j}^* \frac{\sum_{d\in D_l}^D \gamma_{d,i}^*}{\sum_{d=1}^D \gamma_{d,i}^t}$$

In the documents where topic $i$ is the largest, $\gamma_{d,i}^t \leq 3\gamma_{d,i}^*$. So, we can conclude

$$\beta_{i,j}^{t+1} \geq (1-\epsilon)\beta_{i,j}^* \frac{1}{3} \frac{\sum_{d\in D_l}^D \gamma_{d,i}^*}{\sum_{d=1}^D \gamma_{d,i}^*}$$

Since $\frac{\sum_{d\in D_l}^D \gamma_{d,i}^*}{\sum_{d=1}^D \gamma_{d,i}^*} \geq (1-o(1))$, as before, we get what we want.

$\square$

**Lemma 42.** *Let the $\beta$ variables have the correct support. Let $j$ be a discriminative word for topic $i$, and let $\beta_{i,j}^t \geq \frac{1}{C_m}\beta_{i,j}^*$, $\gamma_{d,i}^t \geq \frac{1}{C_m}\gamma_{d,i}^*$ whenever $\beta_{i,j}^* \neq 0$, $\gamma_{d,i}^* \neq 0$. Let $\beta_{i,j}^t = C_\beta^t\beta_{i,j}^*$, where $C_\beta^t \geq 4C_m$, and $C_m$ is a constant. Then, in the next iteration, $\beta_{i,j}^{t+1} \leq C_\beta^{t+1}\beta_{i,j}^*$, where $C_\beta^{t+1} \leq \frac{C_\beta^t}{2}$.*

*Proof.* The proof is exactly the same as Lemma 5.

$\square$

Now, we finally get to the upper bound of the $\gamma$ values.

**Lemma 43.** *Fix a particular document $d$. Let's assume the supports for the $\beta$ and $\gamma$ variables are correct. Furthermore, let $\frac{1}{C_m} \leq \frac{\beta_{i,j}^t}{\beta_{i,j}^*} \leq C_m$ for some constant $C_m$. Then, $\gamma_{d,i}^t \leq 2\gamma_{d,i}^*$.*

*Proof.* Again, multiplying 2.1 by $\gamma_{d,i}^t$, we get

$$\gamma_{d,i}^t = \sum_{j\in L_i} \tilde{f}_{d,j} + \gamma_{d,i}^t \sum_{j\notin L_i} \frac{\tilde{f}_{d,j}}{f_{d,j}^t}\beta_{i,j}^t + \gamma_{d,i}^t \sum_{j\in C} \frac{\tilde{f}_{d,j}}{f_{d,j}^t}\beta_{i,j}^t$$

If $\tilde{\alpha} = \sum_{j\in L_i}\beta_{i,j}^*$, since $\gamma_{d,i}^t \geq \frac{1}{C_m}\gamma_{d,i}^*$,

$$\frac{\tilde{f}_{d,j}}{f_{d,j}^t} \leq (1+\epsilon)C_m^2$$

If we denote $\Gamma = \sum_{j\in C}\beta_{i,j}^*$, then

$$\gamma_{d,i}^t \leq (1+\epsilon)(\tilde{\alpha}\gamma_{d,i}^* + C_m^3(1-\Gamma-\tilde{\alpha})\gamma_{d,i}^t + \Gamma\kappa^4\gamma_{d,i}^t)$$

Equivalently, $\gamma_{d,i}^t \leq \frac{(1+\epsilon)\tilde{\alpha}}{1-(1+\epsilon)C_m^3(1-\Gamma-\tilde{\alpha})-(1+\epsilon)\Gamma\kappa^4}\gamma_{d,i}^*$

Then, we claim that $\frac{(1+\epsilon)\tilde{\alpha}}{1-(1+\epsilon)C_m^3(1-\Gamma-\tilde{\alpha})-(1+\epsilon)\Gamma\kappa^4} \leq 1 + \frac{1}{\kappa^{50}}$. Indeed, $\Gamma\kappa^4 \leq \kappa^{-96}$, and $C_m^3(1-\Gamma-\tilde{\alpha}) \leq C_m^3(1-\tilde{\alpha}) = o(1)$. Hence,

$$\frac{(1+\epsilon)\tilde{\alpha}}{1-(1+\epsilon)C_m^3(1-\Gamma-\tilde{\alpha})-(1+\epsilon)\Gamma\kappa^4} \leq \frac{(1+\epsilon)\tilde{\alpha}}{1-o(1)-\kappa^{-96}} \leq \frac{(1+\epsilon)\tilde{\alpha}}{1-\kappa^{-95}}$$

Finally, we claim that $\frac{(1+\epsilon)\tilde{\alpha}}{1-\kappa^{-95}} \leq 1 + \kappa^{-50}$. Indeed, this is equivalent to

$$\tilde{\alpha} \leq (1+\epsilon)(1+\kappa^{-50})(1-\kappa^{-95}) \leq (1+\epsilon)(1+\kappa^{-50})$$

But, since we assume $\kappa \geq 2$, the claim we need follows easily.

$\square$

## 5.2 Phase II of analysis

Finally, we deal with the alternating minimization portion of the argument. How will we deal with the lack of anchor documents? The almost obvious way: if a document has topic $i$ with proportion $1 - \frac{1}{\kappa^{100}}$, it will behave for all purposes like an anchor document, because the dynamic range of word $\beta_{i,j}^*$ is limited, and the contribution from the other topics is not that significant.

Intuitively, we'll show that $\frac{f_{d,j}^*}{f_{d,j}^t} \approx \frac{\beta_{i,j}^*}{\beta_{i,j}^t}$, so that these documents provide a "push" for the value of $\beta_{i,j}^t$ in the correct direction.

**Lemma 44.** *Let's assume that our current iterates $\beta_{i,j}^t$ satisfy $\frac{1}{C_\beta^t} \leq \frac{\beta_{i,j}*}{\beta_{i,j}^t} \leq C_\beta^t$ for $C_\beta^t \geq \frac{1}{(1-\epsilon)^{20}}$. Then, after iterating the $\gamma$ updates to convergence, we will get values $\gamma_{d,i}^t$ that satisfy $(C_\beta^t)^{1/10} \leq \frac{\gamma_{d,i}*}{\gamma_{d,i}^t} \leq (C_\beta^t)^{1/10}$.*

*Proof.* As before, we have that

$$\gamma_{d,i}^t = \sum_{j \in L_i} \tilde{f}_{d,j} + \gamma_{d,i}^t \sum_{j \notin L_i} \frac{\tilde{f}_{d,j}}{f_{d,j}^t} \beta_{i,j}^t$$

Let's denote as $C_\gamma^t = \max_i (\max(\frac{\gamma_{d,i}^*}{\gamma_{d,i}^t}, \frac{\gamma_{d,i}^t}{\gamma_{d,i}^*}))$, and let, as before, assume that $\frac{\gamma_{d,i_0}^t}{\gamma_{d,i_0}^*} = C_\gamma^t$.

By the definition of $C_\gamma^t$,

$$\gamma_{d,i_0}^t = \sum_{j \in L_{i_0}} \tilde{f}_{d,j} + \gamma_{d,i_0}^t \sum_{j \notin L_{i_0}} \frac{\tilde{f}_{d,j}}{f_{d,j}^t} \beta_{i_0,j}^t \leq$$

$$(1 + \epsilon)(\tilde{\alpha} \gamma_{d,i_0}^* + (1 - \tilde{\alpha})(C_\beta^t)^2 (C_\gamma^t)^2 \gamma_{d,i_0}^*)$$

We claim that

$$(1 + \epsilon)(\tilde{\alpha} + (1 - \tilde{\alpha})(C_\beta^t)^2 (C_\gamma^t)^2) \leq (C_\gamma^t)^{1/10} \tag{5.1}$$

which will be a contradiction to the definition of $C_\gamma^t$.

After a little rewriting, 5.1 translates to $\tilde{\alpha} \geq 1 - \frac{\frac{(C_\gamma^t)^{1/10}}{1+\epsilon} - 1}{(C_\beta^t C_\gamma^t)^2 - 1}$. By our assumption on $C_\gamma^t$, $C_\beta^t \leq C_\gamma^{10}$, so the right hand side above is upper bounded by $1 - \frac{\frac{(C_\gamma^t)^{1/10}}{1+\epsilon} - 1}{(C_\gamma^t)^8 - 1}$.

But, Lemma 43 implies that certainly $C_\gamma^t \leq C_\gamma^0$. The function

$$f(c) = \frac{\frac{c^{1/10}}{1+\epsilon} - 1}{c^8 - 1}$$

can be easily seen to be monotonically decreasing on the interval of interest, and hence is lower bounded by $\frac{\frac{(C_\gamma^0)^{1/10}}{1+\epsilon} - 1}{(C_\gamma^0)^8 - 1}$. Since $\tilde{\alpha} = (1 - o(1))(1 - \frac{1}{\kappa^{100}})$ and $C_\gamma^0 \leq 3$, the claim we want is clearly true.

The case where $\frac{\gamma_{d,i_0}^*}{\gamma_{d,i_0}^t} = C_\gamma^t$ is not much more difficult. An analogous calculation as in Lemma 8 gives that to get a contradiction to the definition of $C_\gamma^t$, the condition required is that $1 - \frac{1 - \frac{1}{(1-\epsilon)(C_\gamma^t)^{1/10}}}{1 - \frac{1}{(C_\gamma^t)^8}}$. As before, if $f(c) = \frac{1 - \frac{1}{(1-\epsilon)c^{1/10}}}{1 - c^8}$, it s easy to check that $f(c)$ is monotonically increasing in the interval of interest, so lower bounded by

$$\frac{1 - \frac{1}{(1-\epsilon)(\frac{1}{(1-\epsilon)^{20}})^{1/10}}}{1 - \frac{1}{((\frac{1}{1-\epsilon})^{20})^8}} =$$

$$\frac{1 - (1 - \epsilon)}{1 - (1 - \epsilon)^{160}} \geq \frac{1}{160}$$

But, $\tilde{\alpha} \geq (1 - \frac{1}{\kappa^{100}})(1 - o(1)) \geq 1 - \frac{1}{160}$, so we get what we want.

$\square$

Next, we show the following lemma.

**Lemma 45.** *Suppose at time step $t$, $\frac{1}{C_\gamma^t}\gamma_{d,i}^* \leq \gamma_{d,i}^t \leq C_\gamma^t\gamma_{d,i}^*$ and $\frac{1}{C_\beta^t}\beta_{i,j}^* \leq \beta_{i,j}^t \leq C_\beta^t\beta_{i,j}^*$, such that $C_\gamma^t \leq (C_\beta^t)^{1/10}$ for $C_\beta^t \geq \frac{1}{(1-\epsilon)^{20}}$. Then, at time step $t+1$, $1/C_\beta^{t+1}\beta_{i,j}^* \leq \beta_{i,j}^t \leq C_\beta^{t+1}\beta_{i,j}^*$, where $C_\beta^{t+1} = (C_\beta^t)^{3/4}$*

*Proof.* Let's assume a document $d$ has a dominating topic of proportion at least $1 - 1/\kappa^{100}$.

Then, we claim that $\frac{f_{d,j}^*}{f_{d,j}^t} \geq \frac{1}{(C_\beta^t)^{1/4}}\frac{\beta_{i,j}^*}{\beta_{i,j}^t}$. We will do a sequence of rearrangements to get this condition to a simpler form:

$$\frac{f_{d,j}^*}{f_{d,j}^t} \geq \frac{1}{(C_\beta^t)^{1/4}}\frac{\beta_{i,j}^*}{\beta_{i,j}^t} \Leftrightarrow$$

$$\frac{f_{d,j}^*}{\beta_{i,j}^*} \geq \frac{1}{(C_\beta^t)^{1/4}}\frac{f_{d,j}^t}{\beta_{i,j}^t} \Leftrightarrow$$

$$\gamma_{d,i}^* + \sum_{i'}\gamma_{d,i'}^*\frac{\beta_{i',j}^*}{\beta_{i,j}^*} > \frac{1}{(C_\beta^t)^{1/4}}(\gamma_{d,i}^t + \sum_{i'}\gamma_{d,i'}^t\frac{\beta_{i',j}^t}{\beta_{i,j}^t})$$

Let's upper bound the right hand side by some simpler quantities. We have:

$$\frac{1}{(C_\beta^t)^{1/4}}(\gamma_{d,i}^t + \sum_{i'}\gamma_{d,i'}^t\frac{\beta_{i',j}^t}{\beta_{i,j}^t}) \leq$$

$$\frac{1}{(C_\beta^t)^{1/4}}C_\gamma^t(\gamma_{d,i}^* + \sum_{i'}\gamma_{d,i'}^*\frac{\beta_{i',j}^t}{\beta_{i,j}^t}) \leq$$

$$\frac{1}{(C_\beta^t)^{1/4}}C_\gamma^t(\gamma_{d,i}^* + (C_\beta^t)^2\sum_{i'}\gamma_{d,i'}^*\frac{\beta_{i',j}^*}{\beta_{i,j}^*})$$

Hence, it is sufficient to prove

$$\gamma_{d,i}^* + \sum_{i'}\gamma_{d,i'}^*\frac{\beta_{i',j}^*}{\beta_{i,j}^*} \geq \frac{1}{(C_\beta^t)^{1/4}}C_\gamma^t(\gamma_{d,i}^* + (C_\beta^t)^2\sum_{i'}\gamma_{d,i'}^*\frac{\beta_{i',j}^*}{\beta_{i,j}^*}) \Leftrightarrow$$

$$\gamma_{d,i}^*(1 - \frac{C_\gamma^t}{(C_\beta^t)^{1/4}}) \geq \sum_{i'}\gamma_{d,i'}^*(\frac{C_\gamma^t}{(C_\beta^t)^{1/4}}(C_\beta^t)^2 - 1)\frac{\beta_{i',j}^*}{\beta_{i,j}^*}$$

Again, we can upper bound the right hand side by

$$\sum_{i'}\gamma_{d,i'}^*(\frac{C_\gamma^t}{(C_\beta^t)^{1/4}}(C_\beta^t)^2 - 1)\kappa =$$

$$(1 - \gamma_{d,i}^*)(\frac{C_\gamma^t}{(C_\beta^t)^{1/4}}(C_\beta^t)^2 - 1)\kappa$$

So, it is sufficient to prove:

$$(1 - \gamma_{d,i}^*)(\frac{C_\gamma^t}{(C_\beta^t)^{1/4}}(C_\beta^t)^2 - 1)\kappa \leq \gamma_{d,i}^*(1 - \frac{C_\gamma^t}{(C_\beta^t)^{1/4}}) \Leftrightarrow$$

$$\gamma_{d,i}^*(1 - \frac{C_\gamma^t}{(C_\beta^t)^{1/4}} + (\frac{C_\gamma^t}{(C_\beta^t)^{1/4}}(C_\beta^t)^2 - 1)\kappa) \geq (\frac{C_\gamma^t}{(C_\beta^t)^{1/4}}(C_\beta^t)^2 - 1)\kappa \Leftrightarrow$$

$$\gamma_{d,i}^* \geq 1 - \frac{1 - \frac{C_\gamma^t}{(C_\beta^t)^{1/4}}}{1 - \frac{C_\gamma^t}{(C_\beta^t)^{1/4}} + (\frac{C_\gamma^t}{(C_\beta^t)^{1/4}}(C_\beta^t)^2 - 1)\kappa}$$

It's easy to check that the expression on the right hand side as a function of $C_\gamma^t$ is decreasing. Hence, the RHS is upper bounded by

$$1 - \frac{1 - \frac{1}{(C_\beta^t)^{3/20}}}{1 - \frac{1}{(C_\beta^t)^{3/20}} + \kappa((C_\beta^t)^{37/20} - 1)}$$

Now, let's analyze this expression. If we let $f(x) = 1 - \frac{1 - \frac{1}{x^{3/20}}}{1 - \frac{1}{x^{3/20}} + \kappa(x^{37/20} - 1)}$, I claim $f(x)$ is an increasing function of $x$. Indeed, we can calculate it's derivative fairly easily:

$$f'(x) = -\frac{\frac{3}{20}x^{-\frac{23}{20}}(1 - \frac{1}{x^{3/20}} + \kappa(x^{37/20} - 1)) - (1 - \frac{1}{x^{3/20}})(-\frac{3}{20}x^{-\frac{23}{20}} + \frac{37}{20}\kappa x^{\frac{17}{20}})}{(1 - \frac{1}{x^{3/20}} + \kappa(x^{37/20} - 1))^2} =$$

$$-\frac{\frac{3}{20}x^{-\frac{23}{20}}\kappa(x^{\frac{37}{20}} - 1) - \frac{37}{20}\kappa x^{\frac{17}{20}}(1 - x^{-\frac{3}{20}})}{(1 - \frac{1}{x^{3/20}} + \kappa(x^{37/20} - 1))^2} = \frac{\frac{\kappa}{20}(40x^{14/20} - (3x^{-23/40} + 37x^{17/20}))}{(1 - x^{3/20} + \frac{1}{\kappa}(\frac{1}{x^{37/20}} - 1))^2}$$

By the AM-GM inequality, $3x^{-23/40} + 37x^{17/20} \geq 40((x^{17/20})^3 7(x^{-23/20})^3)^{1/40} = 40x^{14/20}$, so $f'(x)$ is positive, so the RHS, as a function of $C_\beta^t$, is $(x)$ increasing.

So, it is sufficient to satisfy the inequality when $C_\beta^t = C_\beta^0$. One can check however that by Lemma 41 and 42 this is true.

Proceeding to the lower bound, a similar calculation as before gives that the necessary condition for progress is:

$$\gamma_{d,i}^* \geq 1 - \frac{1 - \frac{(C_\beta^t)^{1/4}}{C_\gamma^t}}{1 - \frac{(C_\beta)^{1/4}}{C_\gamma^t} + \frac{1}{\kappa}(\frac{(C_\beta^t)^{1/4}}{C_\gamma^t}\frac{1}{(C_\beta^t)^2} - 1)}$$

Again, the right hand side expression is decreasing in $C_\gamma$, so it is certainly upper bounded by

$$1 - \frac{1 - (C_\beta^t)^{3/20}}{1 - (C_\beta^t)^{3/20} + \frac{1}{\kappa}(\frac{1}{(C_\beta^t)^{37/20}} - 1)}$$

Now, the claim is that this expression is increasing in $C_\beta^t$. Again, denoting $f(x) = 1 - \frac{1 - x^{3/20}}{1 - x^{3/20} + \frac{1}{\kappa}(\frac{1}{x^{37/20}} - 1)}$

$$f'(x) = -\frac{-\frac{3}{20}x^{-17/20}(1 - x^{3/20} + \frac{1}{\kappa}(\frac{1}{x^{37/20}} - 1)) - (1 - x^{3/20})(-\frac{3}{20}x^{-17/20} - \frac{1}{\kappa}\frac{37}{20}x^{-57/20})}{(1 - x^{3/20} + \frac{1}{\kappa}(\frac{1}{x^{37/20}} - 1))^2} =$$

$$-\frac{-\frac{3}{20}x^{-17/20}\frac{1}{\kappa}(\frac{1}{x^{37/20}} - 1) + (1 - x^{3/20})\frac{1}{\kappa}\frac{37}{20}x^{-57/20}}{(1 - x^{3/20} + \frac{1}{\kappa}(\frac{1}{x^{37/20}} - 1))^2} = \frac{\frac{1}{20\kappa}(-40x^{-54/20} + (3x^{-17/40} + 37x^{-57/20}))}{(1 - x^{3/20} + \frac{1}{\kappa}(\frac{1}{x^{37/20}} - 1))^2}$$

By the AM-GM inequality, $3x^{-17/40} + 37x^{-57/20} \geq 40((x^{-17/20})^3 7(x^{-57/20})^{37})^{1/40} = 40x^{-54/20}$, so $f'(x)$ is negative, so the RHS, as a function of $C_\beta^t$, is decreasing. So it suffices to check the inequality when $C_\beta^t = (1 - \epsilon)^{20}$. In this case, we want to check that

$$1 - \frac{1}{\kappa^{100}} \geq 1 - \frac{1 - \frac{1}{(1-\epsilon)^3}}{1 - \frac{1}{(1-\epsilon)^3} + \frac{1}{\kappa}((1 - \epsilon)^{37} - 1)}$$

Since $1 - \frac{1 - \frac{1}{(1-\epsilon)^3}}{1 - \frac{1}{(1-\epsilon)^3} + \frac{1}{\kappa}((1-\epsilon)^{37} - 1)} \leq 1 - \frac{3\kappa}{37 + 3\kappa}$, and $\kappa \geq 2$, this is easily seen to be true.

Now, we'll split the $\beta$ update into two parts: documents where topic $i$ is at least $1 - 1/\kappa^{100}$, and the rest of them. In the first group, as we showed above, $\frac{f_{d,j}^*}{f_{d,j}^t} \geq \frac{1}{(C_\beta^t)^{1/2}}$. In the second group, we can certainly claim that $\frac{f_{d,j}^*}{f_{d,j}^t} \geq \frac{1}{C_\gamma^t C_\beta^t}$ from the inductive hypothesis. If we denote the set of documents where topic $i$ is at least $1 - 1/\kappa^{100}$ as $D_1$, we get that

$$\beta_{i,j}^{t+1} = \beta_{i,j}^t \frac{\sum_d \frac{f_{d,j}^*}{f_{d,j}^t}\gamma_{d,i}^t}{\sum_{i=1}^D \gamma_{d,i}^t} \geq$$

$$\frac{\sum_{d \in D_1} \frac{1}{(C_\beta^t)^{1/2}C_\gamma^t}\beta_{i,j}^*\gamma_{d,i}^* + \sum_{d \in D \setminus D_1} \frac{1}{(C_\beta^t)^2(C_\gamma^t)^2}\beta_{i,j}^*\gamma_{d,i}^*}{(C_\gamma^t)\sum_{d \in D}\gamma_{d,i}^*}$$

If we denote $\mu = \frac{\sum_{d \in D_1}\gamma_{d,i}^*}{\sum_{d \in D}\gamma_{d,i}^*}$, then

$$\beta_{i,j}^{t+1} \geq \mu \frac{\beta_{i,j}^*}{(C_\beta^t)^{1/4}(C_\gamma^t)^2} + (1 - \mu)\frac{\beta_{i,j}^*}{(C_\beta^t)^2(C_\gamma^t)^3}$$

So, to prove $\beta_{i,j}^{t+1} \geq \frac{1}{C_\beta^{3/4}}\beta_{i,j}^*$, it's sufficient to show

$$\mu \frac{\beta_{i,j}^*}{(C_\beta^t)^{1/4}(C_\gamma^t)^2} + (1-\mu)\frac{\beta_{i,j}^*}{(C_\beta^t)^2(C_\gamma^t)^3} \geq \frac{1}{C_\beta^{3/4}} \Leftrightarrow$$

$$\mu > \frac{\frac{1}{(C_\beta^t)^{1/2}} - \frac{1}{(C_\beta^t)^2(C_\gamma^t)^3}}{\frac{1}{(C_\beta^t)^{1/4}(C_\gamma^t)^2} - \frac{1}{(C_\beta^t)^2(C_\gamma^t)^3}}$$

Given that $C_\gamma^t \leq (C_\beta^t)^{1/10}$, it's sufficient to show

$$\mu > \frac{\frac{1}{(C_\beta^t)^{1/2}} - \frac{1}{(C_\beta^t)^{23/10}}}{\frac{1}{(C_\beta^t)^{9/20}} - \frac{1}{(C_\beta^t)^{23/10}}} = 1 - \frac{\frac{1}{(C_\beta^t)^{9/20}} - \frac{1}{(C_\beta^t)^{1/2}}}{\frac{1}{(C_\beta^t)^{9/20}} - \frac{1}{(C_\beta^t)^{23/10}}}$$

Completely analogously as before, $1 - \frac{\frac{1}{(C_\beta^t)^{9/20}} - \frac{1}{(C_\beta^t)^{1/2}}}{\frac{1}{(C_\beta^t)^{9/20}} - \frac{1}{(C_\beta^t)^{23/10}}}$ is a decreasing function of $C_\beta^t$, so it's sufficient

to check that $\mu > 1 - \frac{\frac{1}{(C_\beta^t)^{9/20}} - \frac{1}{(C_\beta^t)^{1/2}}}{\frac{1}{(C_\beta^t)^{9/20}} - \frac{1}{(C_\beta^t)^{23/10}}}$ when $C_\beta^t = (\frac{1}{1-\epsilon})^{20}$, which is easily checked to be true.

In the same way, one can prove that $\beta_{i,j}^{t+1} \leq (C_\beta^t)^{3/4}\beta_{i,j}^*$ $\qquad\qquad \square$

Putting lemmas 44 and 45 together, we get that the analogue of Lemma 10:

**Lemma 46.** *Suppose it holds that* $\frac{1}{C^t} \leq \frac{\beta_{i,j}*}{\beta_{i,j}^t} \leq C^t$, $C^t \geq \frac{1}{(1-\epsilon)^{20}}$. *Then, after one KL minimization step with respect to the* $\gamma$ *variables and one* $\beta$ *iteration, we get new values* $\beta_{i,j}^{t+1}$ *that satisfy* $\frac{1}{C^{t+1}} \leq \frac{\beta_{i,j}*}{\beta_{i,j}^{t+1}} \leq C^{t+1}$, *where* $C^{t+1} = (C^t)^{3/4}$

As a corollary,

**Corollary 47.** *Phase III requires* $O(\log(\frac{1}{\log(1+\epsilon)})) = O(\log(\frac{1}{\epsilon}))$ *iterations to estimate each of the topic-word matrix and document proportion entries to within a multiplicative factor of* $\frac{1}{(1-\epsilon)^7}$

This finished the proof of Theorem 39 for Case Study 1.

## 5.3 Generalizing Case Study 2

Finally, the proof for Case Study 2 is quite simple. Because the dynamic range $\kappa \leq B$ for the common words, Lemmas 33 and 34 still hold, and hence we again determine the dominant topic correctly. Because of this, it's also easy to see that the lower bounds and upper bounds on the $\beta_{i,j}^t$ values for the common words are maintained to be a constant, since the proof of Lemmas 30 and 31 holds for the common words verbatim. This means that the anchor words and discriminative words will be correctly determined just as before. But after that point, the analysis of Case Study 2 is exactly the same as the one for Case Study 2 — which we already covered in the above section. This finishes the proof of Theorem 39.

# 6 Estimates on number of documents

Finally, we state a few helper lemmas to estimate how many documents will be needed. The properties we need are that the empirical marginals of a dominating topic in the documents where it's dominating are close to the actual ones, and similarly that the empirical marginals of the dominating topic, conditioned on the set of topics that a discriminative word belongs to not being present are close to the actual ones.

The former statement is the following:

**Lemma 48.** *Let $E_i = \mathbf{E}[\gamma_{d,i}^* | \gamma_{d,i}^*$ is dominating]. If the total number of documents is $D = \Omega(\frac{K \log^2 K}{\epsilon^2})$, and $D_i$ is the number of documents where $i$ is the dominant topic, then with high probability, for all topics $i$,*

$$(1 - \epsilon)E_i \leq \frac{1}{D_i} \sum_{d \in D_i} \gamma_{d,i}^* \leq (1 + \epsilon)E_i$$

*Proof.* Since documents are generated independently, $\Pr[\frac{1}{D_i} \sum_{d \in D_i} \gamma_{d,i}^* > (1+\epsilon)E_i] \leq e^{-\frac{\epsilon^2 D_i E_i}{3}}$ by Chernoff.

Since there are at most $T$ topics per document, $E_i \geq \frac{1}{T}$, so $\Pr[\frac{1}{D_i} \sum_{d \in D_i} \gamma_{d,i}^* > (1+\epsilon)E_i] \leq e^{-\frac{\epsilon^2 D_i}{3T}}$

An analogous statement holds for $\Pr[\frac{1}{D_i} \sum_{d \in D_i} \gamma_{d,i}^* < (1-\epsilon)E_i]$

Then, if $D_i = \frac{\log^2 K}{\epsilon^2}$, by union bounding, we get that with high probability, for all topics, $(1 - \epsilon)E_i \leq \frac{1}{D_i} \sum_{d \in D_i} \gamma_{d,i}^* \leq (1 + \epsilon)E_i$

However, the probability of a topic being dominating is $C_i/K$ for some constant $C_i$. So, by another Chernoff bound,

$$\Pr[D_i < (1-\epsilon)C_i D/K] \leq e^{-\frac{\epsilon^2 C_i D}{3K}} \tag{6.1}$$

So, if we take $D = \frac{K}{\epsilon^2} \log^2 K$, with high probability, for all topics, $D_i = \Theta(D/K)$.

Putting everything together, we get that if $D = \frac{K \log^2 K}{\epsilon^2}$, with high probability,

$$(1 - \epsilon)E_i \leq \frac{1}{D_i} \sum_{d \in D_i} \gamma_{d,i}^* \leq (1 + \epsilon)E_i$$

$\square$

Next, we calculate how many documents are needed to match the marginals of the dominating topics, conditioned on a small subset (of size $o(K)$) of the topics not being included in a document. More formally,

**Lemma 49.** *For the discriminative word $j$, let $jS$ be the set of topics it belongs to. For a topic $i \in jS$, let Let $E_{i,jS} = \mathbf{E}[\gamma_{d,i}^* | \gamma_{d,i}^*$ is dominating, $\gamma_{d,i'}^* = 0, \forall i' \in jS]$. Let $D_{i,jS}$ be the number of documents where $i$ is dominating, and $\gamma_{d,i'}^* = 0, \forall i' \in jS$.*

*If the number of documents $D \geq \frac{K \log^2 N}{\epsilon^2}$, then with high probability, for all topics $i$ and discriminative words $j$, $(1 - \epsilon)E_{i,jS} \leq \frac{1}{D_{i,jS}} \sum_{d \in D_{i,jS}} \gamma_{d,i}^* \leq (1 + \epsilon)E_{i,jS}$*

*Proof.* Since $E_{i,jS} = (1 \pm o(1))E_i$, by the weak topic correlation property, an analogous proof as above shows that if we get that if $D_{i,jS} = \frac{\log^2 K}{\epsilon^2}$, with high probability, $(1 - \epsilon)E_{iS} \leq \frac{1}{D_{iS}} \sum_{d \in D_{iS}} \gamma_{d,i}^* \leq (1 + \epsilon)E_{iS}$.

But by the independent topic inclusion property, the probability of generating a document $D$ with $i$ being the dominating topic, s.t. no topics in $jS$ appear in it is $\Theta(1/K)$. So, again by Chernoff,

$$\Pr[D_{i,jS} < (1-\epsilon)C_i D/K] \leq e^{-\frac{\epsilon^2 C_i D}{3K}} \tag{6.2}$$

If we take $D = \frac{K}{\epsilon^2} \log^2 N$, $\Pr[D_{i,jS} < (1-\epsilon)C_i D/K] \leq e^{-\log^2 N}$. However, since the total number of $i, jS$ pairs is at most $N^2$, union bounding, we get that with high probability, for all pairs $i, jS$,

$$(1 - \epsilon)E_{i,jS} \leq \frac{1}{D_{i,jS}} \sum_{d \in D_{i,jS}} \gamma_{d,i}^* \leq (1 + \epsilon)E_{i,jS}$$

$\square$

Finally, the following short lemma to estimate the number of documents in which a word $j$ belongs only to the dominating topic is implicit in the proof above:

**Lemma 50.** *Let $D_{i,jS}$ be the number of documents where $i$ is dominating, and $\gamma^*_{d,i'} = 0, \forall i' \in j_S$. If the number of documents $D \geq \frac{K \log^2 N}{\epsilon^2}$, then with high probability, for all topics $i$ and discriminative words $j$, $D_{i,jS} \geq D_i(1 - \epsilon)(1 - o(1))$*

## Footnotes

[1]One gets these trivially, turning the constraint that $\sum_{i=1}^K \gamma_{d,i}^t = 1$ into a Lagrange multiplier

# References

M. Telgarsky. Dirichlet draws are sparse with high probability. Manuscript, 2013.