[Reviews · NeurIPS 2015]

Submitted by Assigned_Reviewer_1

The authors prove that variational inference in LDA converges to the ground truth model, in polynomial time, for two different case studies with different underlying assumptions about the structure of the data. In this analysis, the authors employ "thresholded" EM updates which estimate the per-topic word distribution based on the subset of documents where a given document dominates. The proofs, which are provided in a 35-page supplement, require assumptions about the number of words in a document that are uniquely associated with each topic, the number of topics per document, and the number documents in which a given word exclusively identifies a topic.

I am not enough of a specialist to evaluate the provided proofs in detail, so I will restrict myself to relatively high level comments. Empirically speaking, variational inference can and does get stuck in local maxima. This suggests that at least some of the assumptions that the authors make will not (always) hold for real-world datasets. As such my intuition is that the authors may have identified a set of conditions under which variational inference in LDA is provably "easy". This in itself is a potentially useful contribution. Assuming the proofs are sound, I would argue for acceptance.

## Comments on content

- I'm a bit confused about the role of the  parameters, about which the

authors appear to make several conflicting statements

> The  Dirichlet parameters do not have a closed form expression and are

> updated via gradient descent.

Do the authors in fact perform emprical Bayes / type II max likelihood

estimation? The statement below suggests otherwise

> In particular, in the E-step, in the large document limit, the first

> term in the update equation for  has a vanishing contribution

The authors further on state that

> These assumptions are a less smooth version of properties of the

> Dirichlet prior. Namely, it's a folklore result that Dirichlet draws

> are sparse with high probability, for a certain reasonable range of

> parameters.

Dirichlet draws are indeed sparse when  << 1.0, but not necessarily when

 => 1.0. This is not so much a "folklore result" as a well known

property of the Dirichlet distribution. Indeed, the lemma referenced in

the supplementary material assumes _i = C_i / K^c with c > 1.

The authors assume  can be ignored in EM updates, but at the same time

make assumptions about sparsity. It would appear they are implicitly

making assumptions about the  true generative process of the data, even

when the same values for  are not necessarily used during inference. Can

they comment on this?

- What is meant by "if i does not belong to document d" in Algorithm 1?

- The exposition in section 4.2 as a whole is somewhat confusing. I would

recommend that the authors simply write down their proposed updates, then

explain how these updates relate to the ones in [Lee and Seung 2000], and

then argue what guarantees apply to these updates.

## Comments on style

- It would benefit the readability of this paper if the authors would put

equations on separate lines (and number them) -- at the very least in those

cases where equations require a summation sign. This also eliminates

awkward back references like "the first term in the update equation for ".

- Similarly, explicitly writing out the generative model, even if well known,

never hurts. It avoids vague phrases like "sampled according to a prior

distribution ".

- There is some confusion of notation between X and w, which are both used to

refer to the words in the document set.

- There is also some confusion as to the indexing of variables, which in some

cases do and in others do not include the document index d.

- This is a matter of personal preference, but I would use `discrete` rather

than `multinomial` when describing the generative model of LDA.

- The words "latent Dirichlet allocation" do not appear in the manuscript.

Why did the authors consistently write "topic model" to refer to LDA?

## Typos

- 131: min -> \min
Summary: The authors present proofs of polynomial time convergence of variational inference in LDA for two different case studies with different underlying assumptions about the structure of the data. The work seems thorough,

although the reviewer was not able to evaluate the proofs in the 35-page supplement in detail.

Submitted by Assigned_Reviewer_2

SUMMARY

This paper shows that under certain conditions, and with suitable initialization, variational inference is able to correctly recover a topic model.

There are a lot of moving parts:

1. Three algorithms are studied, all simplifications of the usual variational scheme.

2. For initialization, it is assumed that for each topic, the user has supplied a document that is an exemplar of that topic (that is, contains that topic with significant weight).

3. There are various assumptions on the topic-word matrix, for instance: -- each word appears in a o(1) fraction of the topics -- different topics have word distributions whose supports are almost disjoint

4. There are also assumptions on the generative process, for instance: -- Each document has a dominating topic: one whose weight exceeds all others by a constant -- Each topic is equally likely to be the dominating topic of a document -- Topics are uncorrelated, more or less

Under these conditions, variational inference can be shown to recover the topic-word matrix and the topic proportions (of each document) within an arbitrarily tight multiplicative factor. The proof analyzes how the estimates behave in three separate phases, each taking a logarithmic number of rounds (in the number of words and other relevant size parameters).

COMMENTS

This is a technically impressive piece of work. Analyzing a heuristic like variational inference is a truly daunting task, and it is quite remarkable that the authors have found a combination of assumptions, initializations, and analysis techniques that work out.

The remaining comments are intended for the authors. Much as I like this paper, I wish it had a simpler narrative. It is a bit bewildering to keep in mind the many assumptions and many algorithms. Perhaps the three different phases of the analysis are subproblems in their own right, and can be defined as such (perhaps with noiseless and noisy variants)? This would at least make it clear which assumptions are needed when.
Summary: Highly nontrivial results proving that under certain conditions, variational inference correctly identifies topic models.

Submitted by Assigned_Reviewer_3

Light Review: The paper considers showing that variational inference

is a provably correct algorithm for some graphical models. This is a

very interesting question with few positive results of this kind.

(There has been a lot of recent work showing how spectral, tensor and

other methods can be used for provably correct learning for some

models). However, given the short time I had, I couldn't see how

interesting the technical details were and how general the approach.

Summary: See below

Submitted by Assigned_Reviewer_4

This paper shows that under certain initializations, the variational inference algorithms for topic models converges to global optimum.

The intuitions behind the theoretical assumptions are well explained. The main theorems, relating the number of updates and the accuracy of recovery, are interesting and appearing practically informative. Little is discussed on how this result relates existing theories and empirical findings on topic models, in particular, about sparsity. It would be desirable to provide a simple simulation study to validate the theory.

- It is not mentioned if the number of topics, K, is known by the learner.

- Section 4.1: It is known that the result of variational inference for topic models depends on the Dirichlet parameters.

I wonder how the main theorems relate to this phenomenon since the Dirichlet parameters disappear from the simplified updates presented in this section.
Summary: The main theorems are interesting and appearing practically informative. Little is discussed on how this result relates existing theories and empirical findings on topic models, in particular, about sparsity.

Author Feedback
Author rebuttal: We're very grateful for the helpful feedback!
First some clarifications relevant for all reviewers. We'll abbreviate variational inference by VI.

a)A remark about nomenclature raised by Reviewer 1: we use "topic models" instead of "LDA" because our theorems apply quite a bit more generally than LDA:
they apply to any topic model satisfying the conditions in Sections 6 & 7 on the topic-word matrix and topic priors - topic priors need not be Dirichlet.

b)Reviewer 1 had a concern for whether we identified "easy" instances for VI, and whether this is a significant contribution.
We reiterate that our assumptions on the ground truth are qualitatively all very related to ones in prior works on topic models, though they might be quantitatively stronger. (e.g. in terms of proportion of anchor words, what the length of the documents needs to be, etc.) Assumptions of this flavor have been found to be reasonable on real-world data by other researchers. (e.g. properties akin to anchor words/dominating topics have been explored in (Bansal et al. NIPS '14); see Sec. 3)

Given that *no* theoretical claims for iterative heuristics related to EM/VI have been proven - we feel strengthening the assumptions quantitatively is still a valuable first step towards understanding when these methods might work - whence when they should be used in practice, and when methods like Monte Carlo should be preferred. (Note, even with all our assumptions, the proofs of correctness were
quite heavy.)

c)Why are there no experimental results (Reviewer 3 & 4)? As mentioned above, qualitatively the assumptions we make had been verified experimentally before, so it would have made little sense to do it again. We included in Sec. 9 a discussion about removing some of the assumptions (qualitatively or quantitatively). It is definitely possible to get stuck in a local optimum if the dynamic range of the words is large compared to the proportion of anchor words or the topic priors are correlated.
However the instances were engineered to break the method and likely don't reflect real-world instances.
It was unclear to us how to make a "systematic" synthetic data set.

Individual reviewer replies follow.

Reviewer 1:

We clarify the exposition of section 4, concretely alpha parameters and sparsity.
-- As we stated, the theorem statements imply our topic priors need not be Dirichlet. They *do* need to be sparse though. (The "sparse and gapped documents" assumption in Sections 6 & 7.)
The updates will not have variables corresponding to the alpha variables indeed (Algorithms 1,2 & 3). The fact that the alpha variables are dropped can either be taken at face value (i.e. they're a modification of the usual updates, and we prove that this modification does, in fact, work) - or a more intuitive reason is provided in section 4.1: for long documents, the part of the updates involving the alpha variables has negligible contribution.
The statement ("The alpha Dirichlet parameters do not have a closed form expression...") refers to the usual VI
updates (Blei et al.). We start section 4 by reviewing these first, as we thought this is what most readers are familiar with.

--"i belongs to document d" means the initialization algorithm returns a support of document d that includes topic i.

-- X and w are used interchangeably in the beginning of section 4, as initially we explain how EM/VI works for general latent variable models, and in that exposition X is a general observable. For topic models, the observables are the words w.

Reviewer 2: Thank you for the kind words and suggestions on improving the readibility!

Reviewer 3: We're unsure what existing theories the reviewer is referring to. The paper pointed out seems at best tangentially related - could the reviewer elaborate more? It considers what the implied sparsity is of the *optimal* variational solution vs the one returned by the MAP solution, for an appropriate definition of sparsity.
It doesn't consider at all parameter learning (via any updates), and is completely asymptotic, in all relevant model quantities. For us, the documents are sparse in the ground truth; one initialization recovers the supports before running our updates; with the other, the updates eventually "find" the correct (sparse) supports themselves.

Reviewer 4: We note the spectral/tensor methods the reviewer mentioned also started with the study of topic models. (By now some tensor methods indeed apply to a wide variety of latent variable methods, but also sometimes one can get better results for more restricted classes.)
It's plausible that using related "separability" assumptions like anchor words, results here could generalize to wider classes of latent variable models. (We hope people will consider such generalizations.)

Reviewer 5: See c).

Reviewer 6: As mentioned, we get results for topic models beyond LDA. We don't ever advertise analyzing variational inference in other settings.